# Coordination of capsule assembly and cell wall biosynthesis in *Staphylococcus aureus*

Marvin Rausch [1,2], Julia P. Deisinger[1,2], Hannah Ulm[1], Anna Müller [1], Wenjin Li[3], Patrick Hardt[1],
Xiaogang Wang[4], Xue Li[4], Marc Sylvester [5], Marianne Engeser [6], Waldemar Vollmer[7], Christa E. Müller [3],
Hans Georg Sahl[8], Jean Claire Lee[4] & Tanja Schneider[1,2]

The Gram-positive cell wall consists of peptidoglycan functionalized with anionic glycopo-lymers, such as wall teichoic acid and capsular polysaccharide (CP). How the different cell wall polymers are assembled in a coordinated fashion is not fully understood. Here, we reconstitute *Staphylococcus aureus* CP biosynthesis and elucidate its interplay with the cell wall biosynthetic machinery. We show that the CapAB tyrosine kinase complex controls multiple enzymatic checkpoints through reversible phosphorylation to modulate the con-sumption of essential precursors that are also used in peptidoglycan biosynthesis. In addition, the CapA1 activator protein interacts with and cleaves lipid-linked CP precursors, releasing the essential lipid carrier undecaprenyl-phosphate. We further provide biochemical evidence that the subsequent attachment of CP is achieved by LcpC, a member of the LytR-CpsA-Psr protein family, using the peptidoglycan precursor native lipid II as acceptor substrate. The Ser/Thr kinase PknB, which can sense cellular lipid II levels, negatively controls CP synthesis. Our work sheds light on the integration of CP biosynthesis into the multi-component Gram-positive cell wall.

[1] Pharmaceutical Microbiology, University of Bonn, Bonn 53115, Germany. [2] German Center for Infection Research (DZIF), partner site Bonn-Cologne, Bonn 53115, Germany. [3] Pharma Center Bonn, Pharmaceutical Institute, Pharmaceutical Chemistry I, University of Bonn, Bonn 53121, Germany. [4] Division of Infectious Diseases, Department of Medicine, Brigham and Women's Hospital, Harvard Medical School, Boston 02115 MA, USA. [5] Institute of Biochemistry and Molecular Biology, University of Bonn, Bonn 53115, Germany. [6] Kekulé Institute for Organic Chemistry and Biochemistry, University of Bonn, 53121 Bonn, Germany. [7] Center for Bacterial Cell Biology, Medical School, Newcastle University, Newcastle upon Tyne NE2 4AX, UK. [8] Institute of Medical Microbiology, Immunology and Parasitology, University of Bonn, Bonn 53127, Germany. These authors contributed equally: Marvin Rausch, Julia P. Deisinger, Hannah Ulm. Correspondence and requests for materials should be addressed to T.S. (email: tschneider@uni-bonn.de)

The bacterial cell envelope is a complex multilayered structure consisting of peptidoglycan (PG), which in Gram-positive bacteria is densely decorated with glyco-polymers such as wall teichoic acid (WTA) and capsular poly-saccharide (CP). The coordinated synthesis and assembly of these polymers is pivotal for maintenance of cell wall architecture and function[1]. In contrast to the biosyntheses of PG and WTA, for which the individual enzymatic reactions have already been characterized in vitro[2,3], the biochemistry under-lying capsule formation in Gram-positive bacteria is not well understood. Even more so, it is largely unknown how the dif-ferent cell wall synthesis pathways, which share building blocks and membrane carriers, function in a coordinated and integrated fashion.

In the case of Staphylococcus aureus, an important opportu-nistic pathogen[4], the expression of a polysaccharide capsule contributes substantially to the ability to cause invasive disease[5–7]. Serotype 5 and 8 capsular polysaccharide (CP5 and CP8) types are dominant among clinical isolates[6]. S. aureus USA300, which is prevalent in the United States, lacks a capsule due to several conserved mutations within the cap5 locus[8]. However, the majority of USA300-associated infections involved superficial wounds or abscesses[9], and USA300 isolates are not common outside of North America[10]. Among predominant methicillin-resistant S. aureus clones worldwide are the CP8 + lineages ST1, ST30, ST59, ST80, and ST239 and the CP5 + lineages ST5 and ST22.

CP5 and CP8 share similar trisaccharide repeating units, which are identical in monosaccharide composition and sequence, only differing in the glycosidic linkages between the sugars and the sites of O-acetylation[11].

The CP5 biosynthetic gene cluster comprises 16 genes (cap5A–cap5P; Fig. 1a)[12] encoding for proteins involved in polymer biosynthesis[7,13,14], acetylation[15], transport, and the regulation of CP production[16,17]. Database homology searches with amino acid sequences of cap5 operon gene products allowed for the prediction of individual enzymatic functions and the proposal of a pathway for capsule (CP) biosynthesis in S. aureus[18].

Within this pathway (Fig. 1b), synthesis of the soluble building blocks occurs in the cytoplasm via three distinct reaction cascades, through which the universal cell envelope substrate UDP-D-N-acetylglucosamine (UDP-D-GlcNAc), is converted into the three different nucleotide-coupled sugars UDP-N-acetyl-D-fucosamine (UDP-D-FucNAc), UDP-N-acetyl-L-fucosamine (UDP-L-FucNAc) and UDP-N-acetyl-D-mannosaminuronic acid (UDP-D-ManNAcA). The synthesis of the first soluble precursor UDP-D-FucNAc is allegedly catalyzed in a two-step process by the enzymes CapD and CapN. Only recently, the integral mem-brane protein CapD was shown to function as a 4,6-dehydratase, which generates the intermediate UDP-2-acetamido-2,6-dideoxy-D-xylo−4-hexulose[19], proposed to be further converted to UDP-D-FucNAc by the action of the membrane-associated reductase CapN, though experimental evidence is lacking so far. Subse-quently, CapM is supposed to transfer the phosphosugar moiety

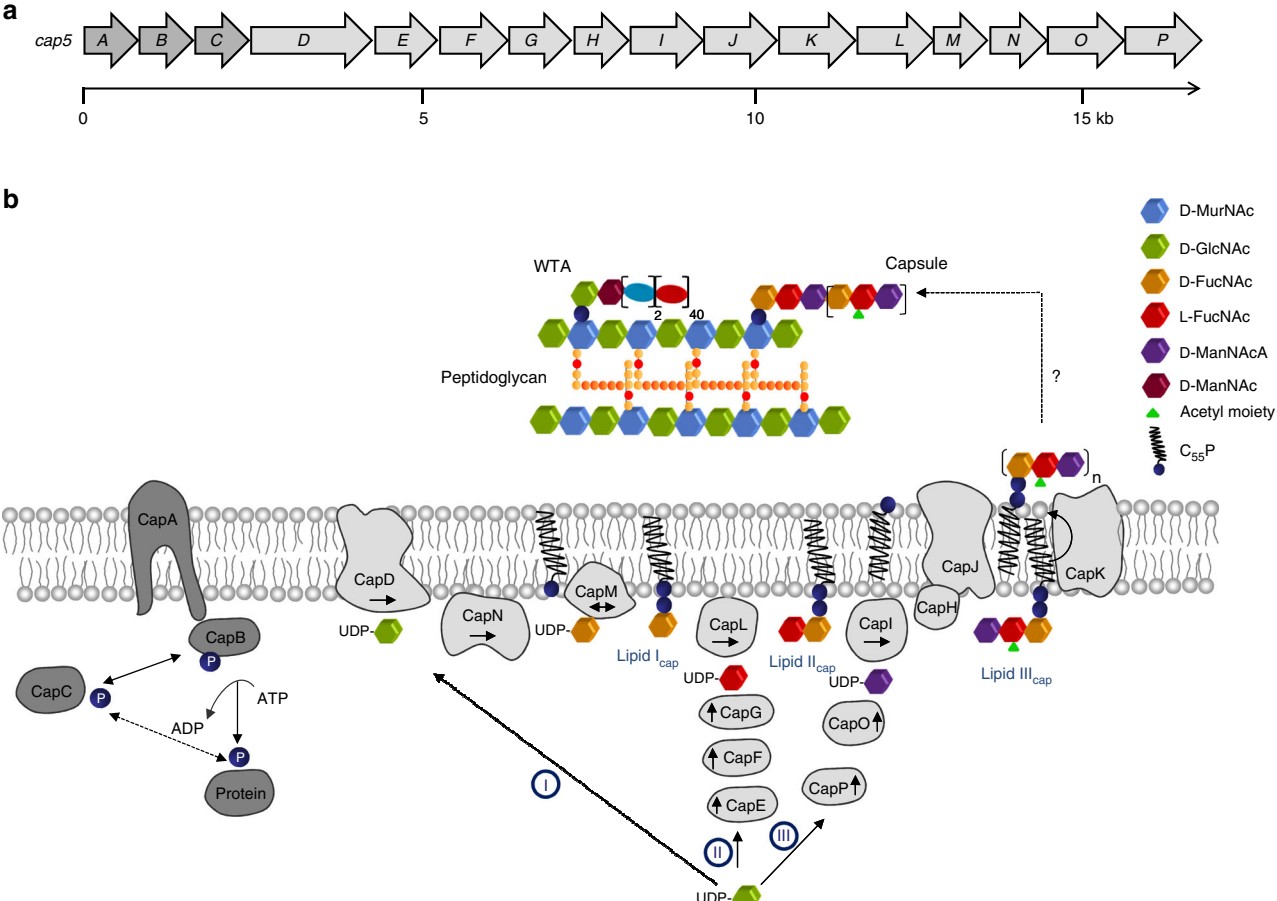

**Fig. 1** S. aureus capsule biosynthesis. **a** S. aureus capsule (CP5) biosynthesis gene cluster (NWMN_0095–0110). **b** Model for the capsule (CP) biosynthesis pathway in S. aureus and its regulation by the tyrosine kinase complex CapA1B1. C$_{55}$P, undecaprenyl-phosphate; GlcNAc, N-acetyl-glucosamine; FucNAc, N-acetyl-fucosamine; ManNAc, N-acetyl-mannosamine; ManNAcA, N-acetyl-mannosaminuronic acid. Arrows indicate synthesis direction. Double arrow indicates reaction reversibility

of UDP-D-FucNAc to the membrane-anchored lipid carrier undecaprenyl-phosphate ($C_{55}P$), yielding lipid $I_{cap}$.

The second cytoplasmic reaction cascade generating the soluble precursor UDP-L-FucNAc involves the enzymes CapE, CapF and CapG, the enzymatic functions of which have already been elucidated biochemically[13]. The transferase CapL is assumed to further attach L-FucNAc to lipid $I_{cap}$ leading to the formation of the second CP lipid intermediate, lipid $II_{cap}$.

The third nucleotide-activated monosaccharide required for CP5 production, UDP-D-ManNAcA, is generated by the epimerase CapP and the dehydrogenase CapO[7,14]. The transmembrane protein CapI has been proposed to transfer the ManNAcA moiety to lipid $II_{cap}$, thereby generating the final capsule precursor lipid $III_{cap}$. The $C_{55}P$-coupled trisaccharide is most likely further modified by the putative acetyltransferase CapH[15], which catalyzes the O-acetylation of L-FucNAc residues in position C3 in CP5 strains[11]. The complete, modified precursor is then translocated to the outer surface of the cell membrane, where polymerization is assumed to take place. These processes are proposed to be facilitated by the putative flippase CapK and the polymerase CapJ, respectively[12,18]. The attachment of CP precursors to the MurNAc (N-acetylmuramic acid) moiety of peptidoglycan is achieved by a yet unknown mechanism possibly involving a member of the LCP (LytR-CpsA-Psr) family of proteins[20,21]. This process likely releases the lipid carrier $C_{55}P$, which enters new synthesis cycles.

The fact that the undecaprenyl-phosphate carrier is found in limited amounts within the cell[22] and required for the biosyntheses of diverse cell envelope components, like CP, WTA and PG[2,3,18], makes a well-orchestrated spatial and temporal regulation of these processes crucial for the viability of the cell. The consequences arising from perturbation of this balanced biosynthetic network have been well exemplified for WTA biosynthesis. Interference with late WTA biosynthesis steps has been shown to be lethal, although the polymer per se is not crucial for viability[23]. Similarly, late stage genes are conditionally essential since they are dispensable for viability in an early gene (tarO or tarA) deletion background, a phenomenon referred to as the "essential gene paradox"[24]. Inhibition of late WTA biosynthesis steps causes the accumulation of dead-end lipid-linked intermediates and thus depletes the cellular pool of $C_{55}P$ to critical levels impeding peptidoglycan biosynthesis, resulting in cell death[25].

In S. aureus, the regulation of CP biosynthesis is not only achieved by differential gene expression[26–28], but has additionally been linked to tyrosine phosphorylation[16,17]. Bacterial tyrosine kinases (BY-kinases) are widespread in bacteria and have multifaceted roles in bacterial exopolysaccharide production[29]. BY-kinases belong to the family of P-loop containing kinases[30], whereby "P-loop" designates a characteristic amino acid sequence resembling the Walker A nucleotide binding motif. BY-kinases of Firmicutes are composed of two interacting polypeptides, a transmembrane activator protein and a cytoplasmic BY-kinase[31]. The cytoplasmic kinase carries a C-terminal tyrosine cluster that undergoes autophosphorylation in the presence of ATP. Based on studies in S. aureus and Streptococcus pneumoniae, the cytoplasmic kinase protein alone is not sufficient for phosphotransfer, but has to interact with the C-terminus of the transmembrane adaptor to undergo autophosphorylation[16,32,33]. Concomitantly, phosphate groups can be transferred to tyrosine residues of target proteins, thus modulating their activity. Two triplets of adjacent genes encoding for a transmembrane adaptor, a cytoplasmic BY-kinase, and a cognate phosphotyrosine phosphatase were identified in the genome of S. aureus serotype 5: The capA1/capB1/capC1 triplet (also referred to cap5A/cap5B/cap5C) is located at the 5′-end of the cap5 operon (Fig. 1a), whereas the highly similar capA2/capB2/capC2 triplet

is found elsewhere on the bacterial chromosome[12,16]. So far, the cellular roles of the distinct CapAB complexes are not fully understood. Particularly, the nature of the stimulus that triggers CapAB signaling and the exact mode of signal transduction are still elusive.

In this study, we functionally reconstitute the entire CP biosynthetic reaction cascade generating the three membrane-anchored CP precursors lipid $I_{cap}$, lipid $II_{cap}$ and lipid $III_{cap}$, allowing us to identify crucial enzymatic check points, which are regulated by the tyrosine kinase CapA1B1 complex to control the consumption of essential precursors. Reconstitution of the membrane anchored CapA1 adaptor protein further reveals a yet elusive function. We show that CapA1 is a dual-function kinase activator/phosphodiesterase protein crucial for signaling and processing of the CP polymer. CapA1 interacts with lipid-bound CP precursors to catalyze the cleavage of the pyrophosphate linkage, releasing the essential lipid carrier $C_{55}P$. Moreover, we elucidate the principles of CP attachment to murein precursors in Gram-positive bacteria. We show that the transfer of the capsular phosphosugar moiety is conducted by a member of the LCP protein family and identify the acceptor substrate. We provide biochemical evidence that the attachment of the anionic precursor likely occurs on the level of the lipid-linked peptidoglycan precursor lipid II, and is facilitated in the presence of CapA1, indicating its cooperative functioning with LCP proteins.

## Results

**In vitro reconstitution of capsule biosynthesis.** Homology searches with CapD and CapN suggested that both proteins are involved in the synthesis of the first soluble capsule precursor UDP-D-FucNAc[18]. More recently, CapD was characterized at the molecular level, and the enzymatic product was shown to be UDP-2-acetamido-2,6-dideoxy-D-xylo−4-hexulose[19]. CapN is proposed to further convert the CapD reaction product to UDP-D-FucNAc by stereospecific reduction of the C-4 keto group[18].

As confirmed by capillary electrophoresis (CE) and mass spectrometry (MS), purified CapN (Supplementary Figure 1) catalyzed the NADPH-dependent conversion of the intermediate UDP-2-acetamido-2,6-dideoxy-D-xylo−4-hexulose to a sugar nucleotide species having a molecular mass of $m/z$ 590.4 for the negatively charged molecule, consistent with the formation of UDP-D-FucNAc (Supplementary Figure 2). The subsequent transfer of the phosphosugar moiety of UDP-D-FucNAc to the lipid anchor $C_{55}P$ is thought to be catalyzed by the polyprenyl-phosphoglycosyltransferase CapM, to initiate the assembly of the lipid-anchored trisaccharide repeating units[18]. Incubation of purified CapM protein with $C_{55}P$ and UDP-D-[$^{14}$C]FucNAc revealed the formation of a new radiolabeled lipid species, not present in the negative control (Fig. 2a). In comparison to $C_{55}P$ (Fig. 2a, lane 1; Rf = 0.95), migration of the lipid product was retarded (lane 3; Rf = 0.87), consistent with the addition of a sugar moiety. The product displayed the same $R_f$ value as determined for the WTA lipid intermediate undecaprenyl-pyrophosphoryl-D-GlcNAc (lipid III), which is structurally very similar to the proposed first CP lipid intermediate[34]. The newly formed purified lipid intermediate had a mass of $m/z$ 1112.7058 for the negatively charged molecule (Fig. 2b), which is consistent with the calculated neutral mass for undecaprenyl-pyrophosphoryl-D-FucNAc (lipid $I_{cap}$) of 1113.7162. Moreover, the GT-B type[35] glycosyltransferase CapL was able to use lipid $I_{cap}$ as acceptor substrate and to catalyze the addition of the second $^{14}$C-labeled sugar moiety L-FucNAc, yielding lipid $II_{cap}$ (Fig. 2a, lane 4; Rf = 0.83). Mass spectrometry analysis of lipid $II_{cap}$ revealed a mass of $m/z$ 1299.7896 [M–H]$^-$ matching the

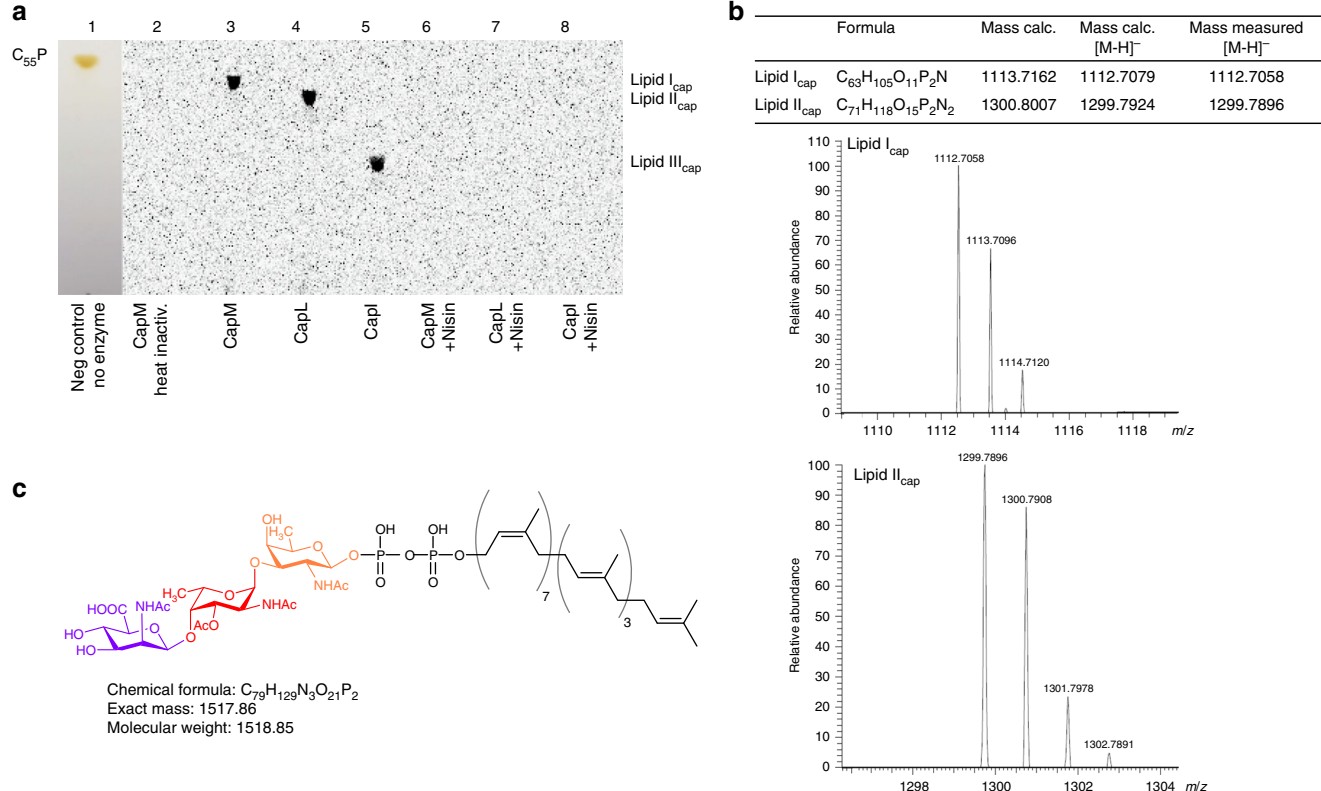

**Fig. 2** Synthesis of CP lipid intermediates. Glycosyltransferases (GT) CapM, CapL and CapI catalyze the in vitro synthesis of CP lipid intermediate lipid I$_{cap}$ (C$_{55}$PP-D-FucNAc), lipid II$_{cap}$ (C$_{55}$PP-D-FucNAc-L-FucNAc) and lipid III$_{cap}$ (C$_{55}$PP-D-FucNAc-L-FucNAc-D-ManNAcA), respectively. **a** Purified GTs were incubated with the respective purified acceptor lipid substrate in the presence of radiolabeled UDP-activated sugars. After extraction with BuOH/PyrAc, synthesis products were analyzed by TLC and phosphoimaging. Counterstain with iodide (lane 1) was used to visualize the migration behavior relative to C$_{55}$P. Nisin, known to bind to the pyrophosphate-sugar moiety of cell wall lipid intermediates was added posterior and formed complexes with all lipid$_{cap}$ intermediates that are not extracted from the reaction mixture. Heat-inactivated CapM was used as a negative control. **b** ESI-MS-analysis of lipid I$_{cap}$ and lipid II$_{cap}$. The peaks at $m/z$ 1112.70 and 1299.78 correspond to the negatively charged lipid I$_{cap}$ and lipid II$_{cap}$ molecules, respectively. **c** Structure of the ultimate CP lipid intermediate lipid III$_{cap}$. Sugar residues are colored: D-FucNAc (orange), L-FucNAc (red), D-ManNAcA (purple)

calculated neutral mass of $m/z$ 1300.801 for undecaprenyl-pyrophosphoryl-D-FucNAc-L-FucNAc.

Conversion to the ultimate lipid-linked CP precursor lipid III$_{cap}$ was achieved by the attachment of a $^{14}$C-D-ManNAcA residue to lipid II$_{cap}$, catalyzed by the glycosyltransferase CapI (Fig. 2a, lane 5; Rf = 0.53). Mass spectrometric analysis of lipid III$_{cap}$ did not yield any signal, presumably due to reduced ionization with addition of further sugar moieties. However, in line with the altered migration behavior of the different CP intermediates on TLC (Fig. 2a, lanes 3–5) resulting from the consecutive sugar residue addition, the lantibiotic nisin formed extraction-stable complexes with all C$_{55}$P-containing CP precursors further validating their identity (Fig. 2a, lanes 6–8). Taken together, these findings confirm the proposed functions of the different glycosyltransferases (GT) and demonstrate that CapM functions as priming GT, initiating the synthesis of membrane-bound CP precursors by coupling the first sugar moiety to the lipid carrier C$_{55}$P.

CapM is homologous to a wide range of GTs from various pathogens (Supplementary Figure 3a), which catalyze the transfer of UDP-activated sugars to C$_{55}$P[36,37]. Prediction of transmembrane topology (http://www.cbs.dtu.dk/services/TMHMM/) and comparison with hydrophobicity plots (ProtScale[38]) of homologous GTs predicts that CapM is anchored to the cytoplasmic membrane by a 28-amino acid long α-helical transmembrane domain that is linked to a large C-terminal catalytic domain located in the cytoplasm (Supplementary Figure 3b). In contrast

to the in silico predicted single-pass transmembrane (TM) helix geometry, structural analysis of the homologous PglC of *Campylobacter concisus* suggests that the TM segment adopts an unusual architecture[37]. Structural modelling of CapM revealed a similar architecture (Supplementary Figure 3c), in which the TM segment is broken into two helices (A and B) with an interhelix angle of 118° by a Ser-Pro motif (Supplementary Figure 3c), suggesting a similar membrane embedment as PglC. Shared structural features further include a conserved Arg residue (Arg3) positioned at the membrane interface as well as a catalytic Asp-Glu dyad and a strictly conserved PRP motif (residues 110–112 in CapM) in the cytoplasmic GT domain, involved in Mg$^{2+}$ and substrate binding, respectively[37] (Supplementary Figure 3). CapM activity in vitro depends on the presence of MgCl$_2$, and was completely lost in the presence of EDTA (Supplementary Figure 4a), as reported for homologous glycosyltransferases[36,37,39].

Interestingly, CapM was inhibited in the presence of tunicamycin (Supplementary Figure 4a), a nucleoside antibiotic shown to inhibit members of the UDP-HexNAc-1-phosphate transferase family (i.e., WecA, MraY or TarO) through competitive binding to the DDxxD Mg$^{2+}$-binding site[39]. Tunicamycin had an IC$_{50}$ of 129 μM for CapM, which is in the range determined for PglC of *C. jejuni* (IC$_{50}$ = of 100 μM)[40]. The lower potency of tunicamycin for these monotopic GTs compared to the polytopic WecA (IC$_{50}$ of 11 nM), MraY (IC$_{50}$ of ~22.5 μM) or TarO (IC$_{50}$ of~59 nM) may likely result from the distinct

architectures and catalytic sites. Since we observed a major impact of anionic phospholipids (dioleoylphosphatidylglycerol and cardiolipin) on the enzymatic activity of CapM, we further investigated the effect of different detergents. However, CapM showed lower activity in the presence of detergents or DMSO indicating that the enzyme preferred the native phospholipid environment (Supplementary Figure 4b).

**CapAB phosphorylation positively modulates CapM and CapE activity.** Full-length integral membrane proteins CapA1 and CapA2 and cytoplasmic kinases CapB1 and CapB2 were over-expressed as $His_6$-tag fusion proteins, purified by Ni-NTA chromatography (Supplementary Figure 1), and utilized to reconstitute CapAB tyrosine kinase activity in vitro in the

presence of γ-labeled [$^{33}$P]ATP. CapB2 autophosphorylation was effectively activated by full-length CapA1 or CapA2 (Fig. 3a) producing a single radiolabeled band with an apparent molecular mass of 25.3 kDa, corresponding to phosphorylated CapB2 (lanes 3 & 4). In contrast, the negative control in which a CapA activator was absent (Fig. 3a, lane 1 and 2, Supplementary Figure 5a) showed no autophosphorylation. In line with previous findings[16], phosphorylation of the individual kinase protein CapB1 was neither observed with CapA1 nor with CapA2 (Fig. 3a, lanes 5 & 6, Supplementary Figure 5a, lanes 6 & 7). However, when CapB1 was fused to its cognate activator CapA1, the recombinant chimera CapAB full-length fusion protein (50.2 kDa) efficiently autophosphorylated in the presence of γ-labeled [$^{33}$P]ATP, demonstrating functionality of CapB1

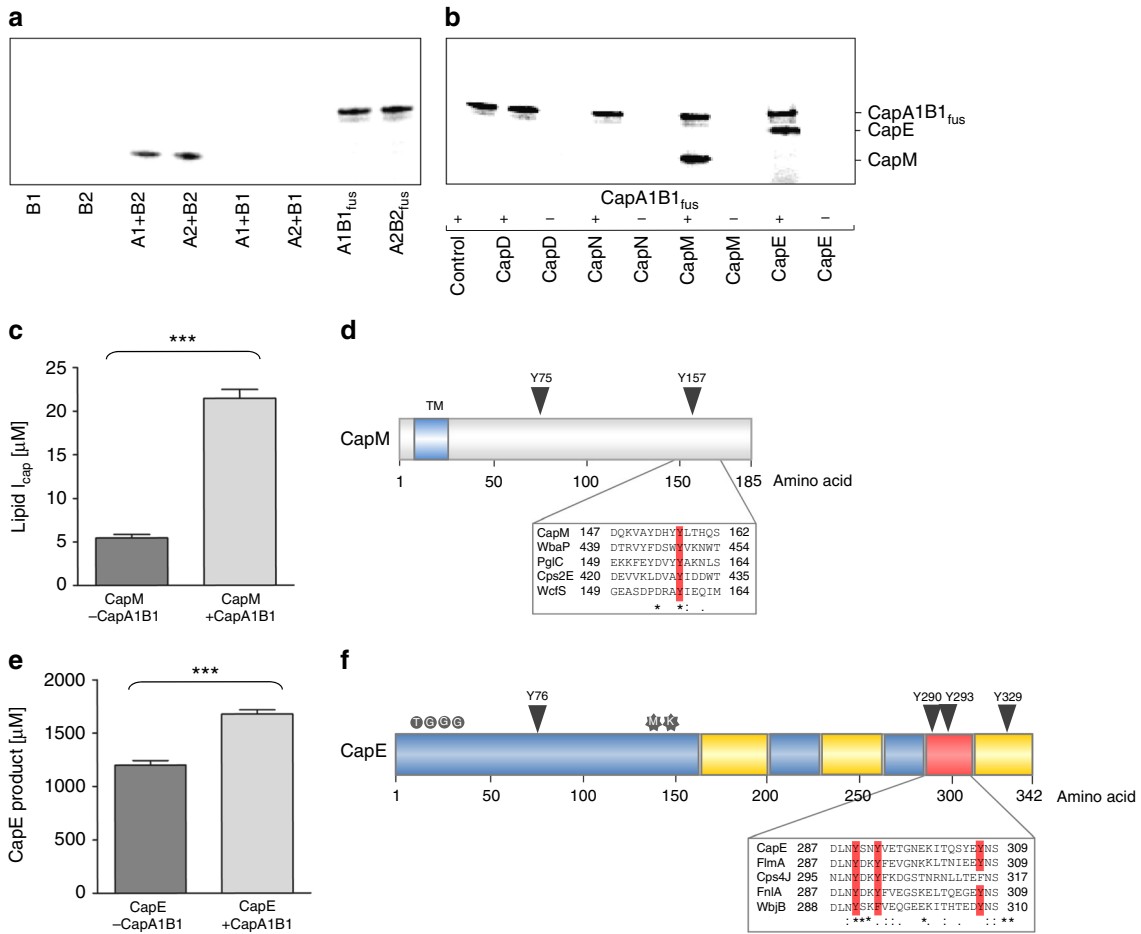

**Fig. 3** Tyrosine phosphorylation positively controls CapM and CapE activity. **a** CapAB autokinase activity was assayed in the presence of [$^{33}$P]ATP using either native proteins or kinase activator CapA1 and tyrosine kinase CapB1 fused into a single polypeptide (CapA1B1$_{fus}$). Phosphotransfer was analyzed by SDS-PAGE and phosphoimaging. **b** CapM and CapE are phosphorylated by the CapA1B1$_{fus}$ tyrosine kinase complex. Putative target proteins (3 µg) were assayed for phosphotransfer in the presence and absence of CapA1B1$_{fus}$. CapA1B1$_{fus}$, 50.2 kDa; CapB2, 25.3 kDa; CapE, 38.6 kDa; CapD, 69.1 kDa; CapM, 21 kDa; CapN, 33.7 kDa. **c** Impact of CapA1B1 kinase activity on CapM glycosyltransferase activity. CapM was incubated in presence of either active CapA1B1 (light grey) or heat-inactivated CapA1B1 (dark grey) with UDP-D-[$^{14}$C]FucNAc and $C_{55}$P. After extraction with BuOH/PyrAc, reactions were analyzed by TLC and phosphoimaging. **d** Topology and CapAB phosphorylation site (arrow) of CapM. The protein is anchored in the membrane by one reentrant transmembrane domain. The catalytic domain is located in the cytoplasm. Sequence alignment of homologous proteins containing the conserved Tyr phosphorylation site (red) is boxed. (from top to bottom: *S. aureus*, *Salmonella enterica*, *Campylobacter concisus*, *Streptococcus pneumoniae* and *Bacteroides fragilis*). **e** Impact of CapA1B1 kinase activity on CapE-mediated substrate conversion. CapE was incubated in presence of either active CapA1B1 (light grey) or heat-inactivated CapA1B1 (dark grey) with UDP-D-GlcNAc. Reactions were analyzed by CE. **f** Mapping of CapE phosphorylation sites. Tyr phosphorylation sites are marked by arrows. Cofactor binding site, substrate-binding site and the mobile loop are marked blue, yellow and red, respectively. The active site motif MxxxK (stars) and the cofactor binding motif TGxxGxxG (circles) are highlighted. A sequence alignment of the mobile latch (red) of selected homologous proteins from different species is boxed (from top to bottom: *S. aureus*, *Bacillus cereus*, *Streptococcus pneumoniae*, *Enterococcus faecium*, *Pseudomonas aeruginosa*). Experiments were performed in triplicate. The error bars represent the ± standard deviation (SD) from three biological replicates. Statistical significance was analyzed by an unpaired *t* test (***$p < 0.005$)

(Fig. 3a, lane 7). A full-length CapA2B2 fusion protein, which was constructed for reasons of comparison, showed comparable autokinase activity (lane 8). This finding contradicts recent suggestions that CapB1 might be a pseudokinase devoid of catalytic activity[41].

CapA1B1-mediated phosphotransfer on putative target proteins involved in CP biosynthesis revealed tyrosine phosphorylation of the glycosyltransferase CapM (21 kDa) and the dehydratase CapE (38.6 kDa) (Fig. 3b, Supplementary Figure 5b). In contrast, recombinant CapD, CapN, CapF, CapG or CapL were not phosphorylated (Fig. 3b, Supplementary Figure 5c). The opposing PHP class phosphatases CapC1 and CapC2 were able to antagonistically dephosphorylate CapB kinase and target proteins (Supplementary Figure 5d).

Phosphorylation of CapM by the CapA1B1 fusion increased lipid $I_{cap}$ synthesis 4-fold (Fig. 3c), showing that CapAB-mediated signaling stimulates the priming step of CP biosynthesis in *S. aureus* by enhancing the rate of CapM-catalyzed glycosyl transfer and thus likely controlling the consumption of the shared carrier $C_{55}P$.

In silico phosphorylation site prediction (NetPhos 3.1)[42] identified tyrosines 75 and 157 as putative phosphosites in CapM (Fig. 3d). The highly conserved Tyr157 appears to be the primary phosphorylation site in CapM, since tyrosine phosphorylation and the concomitant stimulatory effect on the catalytic activity was completely abolished in the CapM_Y157F mutant (Supplementary Figure 6a,b). In contrast, changing Tyr75 to Phe had only a minor effect on the protein's ability to be catalytically activated by CapA1B1-mediated phosphotransfer. Both mutant proteins retained catalytic activity comparable to wild-type CapM in the absence of CapA1B1 (Supplementary Figure 6b) showing that Tyr157 represents the crucial regulatory CapAB phosphorylation site on CapM.

The CapA1B1-mediated stimulation of CapM activity is also reflected by enzyme kinetics, revealing a *Km* value of $2,309 \pm 311.8\,\mu M$ for UDP-D-FucNAc and a $V_{max}$ value of $0.5559 \pm 0.02645$ pmol $min^{-1}\,\mu g^{-1}$ in the absence of CapA1B1 and a lowered $K_m$ value of $996.2 \pm 136.2\,\mu M$ and an increased $V_{max}$ of $1.0688 \pm 0.0358$ pmol $min^{-1}\,\mu g^{-1}$ in the presence of CapA1B1 (Supplementary Figure 6c). Moreover, a CapM phosphomimetic in which Tyr157 was exchanged to Glu displayed a $K_m$ value of $1383 \pm 267.3\,\mu M$ and a $V_{max}$ value of $0.8625 \pm 0.04708$ pmol $min^{-1}\,\mu g^{-1}$ in the absence of CapA1B1, substantiating the stimulatory impact of CapM phosphorylation on Tyr157 (Supplementary Figure 6c).

In vitro kinase assays further identified CapE as a CapA1B1 target (Fig. 3e). Tryptic fragments of in vitro phosphorylated CapE were analyzed by nanoscale liquid chromatography coupled to tandem mass spectrometry (nanoLC-MS/MS), and Tyr76 of the CapE protein was found to be phosphorylated (Supplementary Figure 7a). This residue is located in proximity to the conserved TGxxGxxG motif required for cofactor binding and to the MxxxK catalytic site[43] (Fig. 3f). Three other phosphotyrosine residues (Tyr290, Tyr293 and Tyr329), were mapped to a C-terminal loop of CapE (Fig. 3f; Supplementary Figure 7a). In CapE in vitro assays with active CapAB, the conversion of UDP-D-GlcNAc was increased by 40%, demonstrating that the CapE catalyzed reaction is positively modulated through phosphorylation (Fig. 3e).

The role of the potential regulatory CapAB phosphorylation sites was probed in a site-directed mutagenesis study. As determined by CE quantification, replacing either Tyr76 of CapE or the three C-terminal tyrosine residues Tyr290, Tyr293, and Tyr329 with Phe diminished the activating effect of CapA1B1 on CapE. The stimulatory effect of CapA1B1 on the conversion of UDP-GlcNAc was completely abolished when all four tyrosine residues were mutated to Phe, confirming their role as regulatory phosphorylation sites on CapE (Supplementary Figure 7b).

**CapA1 is a dual-function phosphodiesterase/kinase activator protein.** When purified lipid $I_{cap}$ was incubated in the presence of CapA1, an additional lipid band was detected by TLC analysis (Fig. 4a), which was not visible in the negative control with heat-inactivated CapA1, indicating enzymatic conversion of the first CP lipid precursor. Intriguingly, the reaction product displayed an identical $R_f$ value as the lipid carrier $C_{55}P$. MALDI-TOF MS analysis determined a mass of $m/z$ 845.610 (negative mode; [M–H]−) for the purified lipid product, confirming that CapA1 is able to catalyze the cleavage of the pyrophosphate linkage within the lipid $I_{cap}$ intermediate to release $C_{55}P$ (Fig. 4b). Of note, CapA1 was unable to cleave the peptidoglycan precursor lipid $II_{PG}$, but was able to hydrolyze the WTA precursor lipid III (Supplementary Figure 8). The facts that CapA1 interacts with lipid-linked CP precursors, and exhibits phosphodiesterase activity towards lipid $I_{cap}$, implies additional functions for this protein in CP biosynthesis, polymerization and attachment processes that go beyond being a mere "transmembrane activator". Importantly, no cleavage of lipid $I_{cap}$ or lipid $III_{WTA}$ was observed with the CapA1 paralogue CapA2 (Fig. 4a), revealing functional differences between the two proteins.

CapA1 is anchored to the cytoplasmic membrane via two transmembrane domains flanking an outside loop comprising 130 amino acids (Fig. 4c), which likely represents a dual function sensory/catalytic domain involved in recognition and processing of membrane-bound CP precursors.

As evidenced in previous research, CapA1 is crucial for efficient capsule formation in *S. aureus*[44]. Corroborating these studies, complementation with pCapA1 *in trans* enhanced CP production in the serotype 8 strain MW2 (Fig. 4d). Strain MW2 carries a frameshift mutation in *capA1 (cap5A)* that results in expression of a truncated version (171 aa) of the full-length gene product (222 aa)[45]. In contrast, deletion of *capA2* in *S. aureus* Newman did not affect in vivo CP production (Fig. 4d). Similarly, the deletion of cognate CapB kinases revealed, that *capB1* but not *capB2* is required for CP production (Supplementary Figure 7c). In spite of the functional redundancy and overlapping protein target specificities of CapA1B1 and CapA2B2 observed in our in vitro phosphorylation assays, the two kinase complexes clearly have distinct roles in cell physiology.

In contrast to CapA1, CapA2 is not encoded within the *cap5* operon, and does not exhibit phosphodiesterase activity towards lipid $I_{cap}$, which led us to conclude that CapA2 has a distinct function in the biosynthesis of the bacterial cell envelope. More recently, the *S. aureus* exopolysaccharide poly-*N*-acetyl-β-(1,6)-glucosamine (PNAG; also referred to as polysaccharide intercellular adhesin) has been reported to bind to the CapA2 receptor loop[46], suggesting that PNAG might represent a molecular signal detected by CapA2. We therefore investigated the influence of purified PNAG and CP5 on the autophosphorylation rate of the CapAB complexes. As revealed by in vitro kinase assays, CapA2-induced autophosphorylation of CapB2 was inhibited in the presence of PNAG, as well as in the presence of CP5, in a concentration-dependent manner (Supplementary Figure 9). In contrast, CapA1-induced phosphorylation of CapB2 was not diminished, indicating that the observed modulatory effect of exopolysaccharide molecules is mediated via an interaction with the transmembrane activator protein CapA2, and not via a direct inhibitory interaction with the protein kinase CapB2. This finding further substantiates differential roles for CapA1 and CapA2 in cell envelope biosynthesis.

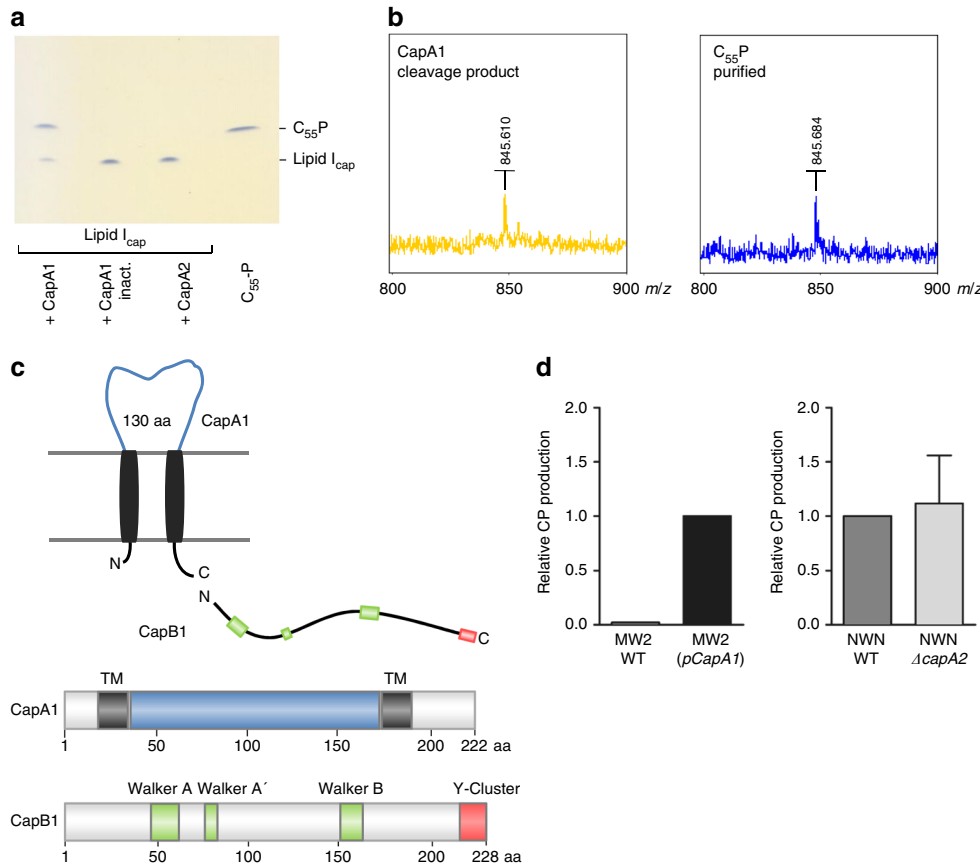

**Fig. 4** CapA1 exhibits phosphodiesterase activity towards lipid-linked capsule precursors. **a** Purified lipid $I_{cap}$ (2 nmol) was incubated in the presence of either CapA1 or CapA2 (4 μg each). After extraction with BuOH/PyrAc, reactions were analyzed by TLC and PMA staining. **b** MS analysis of the CapA1 cleavage product isolated via preparative TLC. MALDI-TOF MS spectra were obtained with a Biflex III instrument running in negative mode. The $m/z$ of 845.61 corresponds to the singly charged $C_{55}P$ molecule, which has a measured mass of 845.68 and a calculated neutral mass of 846.67. **c** CapA1 is anchored to the cytoplasmic membrane via two transmembrane helices flanking an extracytoplasmic loop of 130 amino acids. The catalytic domains of CapB1 kinase contains the Walker A, A′ and B motifs (green); a tyrosine-rich region containing the phosphorylation sites is present at the C-terminus (red). **d** ELISA-based quantification of CP production in *S. aureus* MW2 complemented *in trans* with pCapA1 (left) and *S. aureus* Newman wild type and a *ΔcapA2* deletion mutant (right). Experiments were performed in triplicate. The error bars represent the mean ± SEM from three biological replicates

**LcpC catalyzes the ligation of CP to lipid II**. Structural analysis and phenotypic studies suggest that members of the LCP protein family catalyze the transfer of undecaprenyl-linked intermediates onto the C6-hydroxyl function of *N*-acetylmuramic acid in PG, thereby promoting attachment of WTA and CP in Gram-positive bacteria[20,47]. *S. aureus* encodes three LCP enzymes with semi-redundant functions[21]. As deduced from knockout mutant studies, CP attachment is preferentially catalyzed by LcpC, though LcpA and LcpB may partially compensate for the loss of LcpC[21].

To investigate the proposed role of LcpC in vitro purified [14C] lipid $I_{cap}$ was incubated with purified LcpC protein and lipid $II_{PG}$ as a potential acceptor substrate. In this setup, LcpC was able to catalyze cleavage of the donor substrate lipid $I_{cap}$ and catalyze attachment of the phosphoryl-sugar moiety to the ultimate PG precursor lipid II (Fig. 5a), resulting in a reaction product ($LII_{PG}$—14CCP) that migrates slower on TLC. Identical results were obtained when [14C]-labeled lipid $II_{PG}$ and non-labeled lipid $I_{cap}$ were used as the reaction substrates (Fig. 5b), further verifying the identity of the ligation product. Quantitative analysis revealed that ~50% of the CP lipid precursor was attached to lipid II (Fig. 5c). Strikingly, the LcpC-mediated transferase reaction was significantly enhanced when CapA1 was included in the reaction mixture, indicating cooperative action of the transmembrane activator/phosphodiesterase protein and LcpC (Fig. 5c).

CapA1 alone was unable to catalyze the attachment of the CP precursor (Supplementary Figure 10a). Of note, LcpC was able to catalyze hydrolytic cleavage of lipid $I_{cap}$, but not of lipid $II_{PG}$, a reaction that would likely be deleterious in vivo (Supplementary Figure 10b, c).

Interestingly, PBP2 added posterior to the LcpC reaction efficiently catalyzed the transglycosylation of lipid $II_{PG}$-CP (Fig. 5a, b). The resulting polymerized PBP reaction product is not extracted from the reaction mixture, thus lipid bands vanish from the TLC. The antibiotic moenomycin, known to inhibit the PBP2 catalyzed transglycosylation, fully blocked the polymerization reaction (Fig. 5b). Importantly, LcpC was further able to efficiently transfer the disaccharide of lipid $II_{cap}$ and the trisaccharide of the ultimate CP lipid intermediate lipid $III_{cap}$ to lipid $II_{PG}$, as evidenced by the altered migration of the respective ligation products (Rf = 0.26 and Rf = 0.22, respectively). Efficient processing of ligation products by PBP2 shows that CP attachment could occur in parallel to PG assembly (Fig. 5d).

Interestingly, antisense-RNA mediated depletion of CapA1 in a triple *Δlcp* mutant was lethal, indicating that CapA1 hydrolysis of $C_{55}$P-coupled CP and WTA precursors contributes to rescue *S. aureus* from the accumulation of toxic intermediates (Supplementary Figure 10d) and further supports a functional link between CapA1 and LcpC.

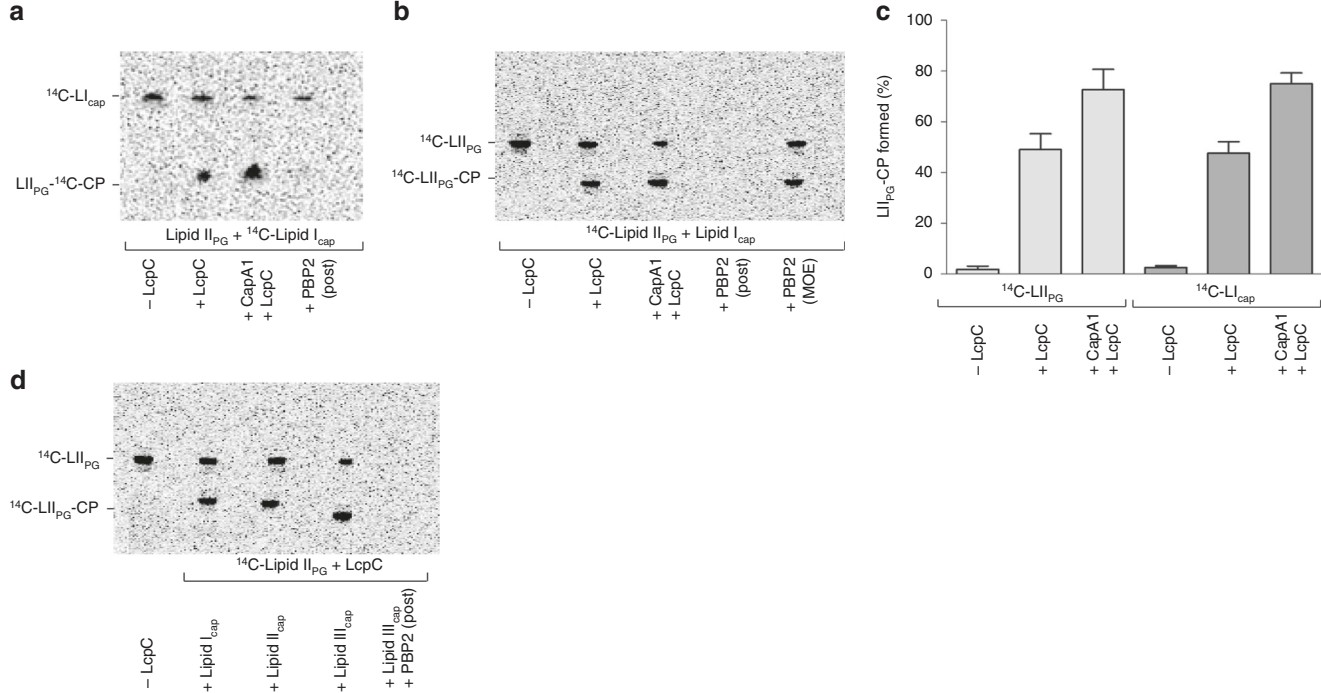

**Fig. 5** Attachment of CP to the ultimate PG building block lipid $II_{PG}$ is mediated by LcpC. **a** [$^{14}$C]lipid $I_{cap}$ and lipid $II_{PG}$ or **b** lipid $I_{cap}$ and [$^{14}$C]lipid $II_{PG}$ were incubated with LcpC in the absence or presence of CapA1. The resulting reaction product ($LII_{PG}$-CP; bold) exhibits an altered migration behavior on the TLC compared to the individual substrates, lipid $II_{PG}$ and lipid $I_{cap}$. PBP2 catalyzes the transglycosylation of the LcpC reaction product $LII_{PG}$-CP (lane 4) resulting in the polymerization of radiolabeled lipid $II_{PG}$-CP into hydrophilic glycan strands that retain in the water phase and are not extracted after BuOH/PyrAc treatment resulting in disappearance of the corresponding band on the chromatogram. The PBP2 catalyzed conversion is inhibited in the presence of the antibiotic moenomycin (MOE; 10 μM) (B, lane 5). **c** Quantification of lipid $I_{cap}$ attachment to lipid $II_{PG}$. Radiolabeled bands were quantified by phosphoimaging. Experiments were performed in triplicate. The error bars represent the ± SD from three biological replicates. **d** LcpC catalyzes the attachment of lipid $II_{cap}$ (lane 3) and the ultimate CP lipid intermediate lipid $III_{cap}$ (lane 4) and the final reaction product is processed by PBP2 added posterior to the LcpC reaction (lane 5). Representative TLC of 2 independent experiments

**Capsule biosynthesis is negatively controlled by PknB**. As evidenced in previous research, the eukaryotic serine/threonine kinase (ESTK) PknB of *S. aureus* is involved in the regulation of cell envelope biogenesis processes[48–50] and was more recently shown to sense lipid $II_{PG}$[51]. Testing CapM in an in vitro kinase assay showed PknB-mediated phosphorylation of the priming GT (Fig. 6a). Moreover, quantitative analysis of the in vitro CapM-catalyzed reaction revealed an up to 30% decrease in lipid $I_{cap}$ synthesis when purified PknB kinase and ATP were included in the reaction mixture, compared to a control reaction containing heat-inactivated PknB, suggesting that ESTK signaling negatively controls the activity of CapM (Fig. 6b). LC/MS-analysis revealed PknB-mediated phosphotransfer onto Thr67, Thr128 and Thr134 of CapM (Supplementary Figure 11). Thr134 is highly conserved among homologous bacterial glycosyltransferases, located in proximity of the membrane interface and the amphipathic helices D and I (Supplementary Figure 3c) and may thus be the most promising candidate for a regulatory PknB-phosphorylation site on CapM.

Intriguingly, the kinase protein CapB1 was also identified as potential phosphorylation target of PknB (Fig. 6c), indicating the possibility of kinase cross-talk. To further explore this finding, the autophosphorylation activity of the CapA1B1 complex was assessed in the presence of PknB. The overall phosphorylation intensity of CapA1B1 was decreased by 34% in the presence of active PknB (Fig. 6d), compared to the heat-inactivated negative control, showing that PknB-mediated serine/threonine phosphorylation inhibits CapB1 autophosphorylation on tyrosine. Thr8 was identified as the specific phosphorylation site on CapB1 by

LC/MS analysis (Supplementary Figure 12a) and a CapA1B1$_{fus}$ phosphomimetic in which this Thr was exchanged to Glu resulted in a significantly reduced CapB1 autophosphorylation (Supplementary Figure 12b, c), suggesting that PknB-mediated phosphorylation interferes with the CapB1 activation by CapA1[41,52]. In contrast, an exchange of Thr8 to Ala did not affect CapA1B1$_{fus}$ autophosphorylation (Supplementary Figure 12b,c).

To determine whether the observed inhibitory effect of PknB in vitro would translate into an in vivo effect, we compared CP5 production in *S. aureus* Newman and in an isogenic *pknB* deletion mutant. Consistent with the proposed role of PknB as negative regulator of CP biosynthesis, the amount of cell-associated CP5 was 5-fold higher in the *pknB* mutant compared to the parental strain and complementation of the Δ*pknB* mutant *in trans* restored CP production to wild-type level (Fig. 6e).

The in vitro and in vivo data reveal that PknB signaling serves to reduce CapM GT activity, as well as CapA1B1 autokinase activity, allowing for a shutdown of CP production likely ensuring a sufficient supply of precursors for PG formation, and thus maintenance of cell wall architecture and function.

## Discussion

Biosynthesis of the *S. aureus* CP shows similarity to the syntheses of PG and WTA, in that all pathways share a pool of essential precursors, i.e. the lipid carrier undecaprenyl-phosphate and UDP-D-GlcNAc[1,53]. Since the availability of $C_{55}P$ within the bacterial cell is limited[22], distribution and prioritization for the alternate metabolic pathways need to be tightly controlled in time and space to ensure bacterial viability.

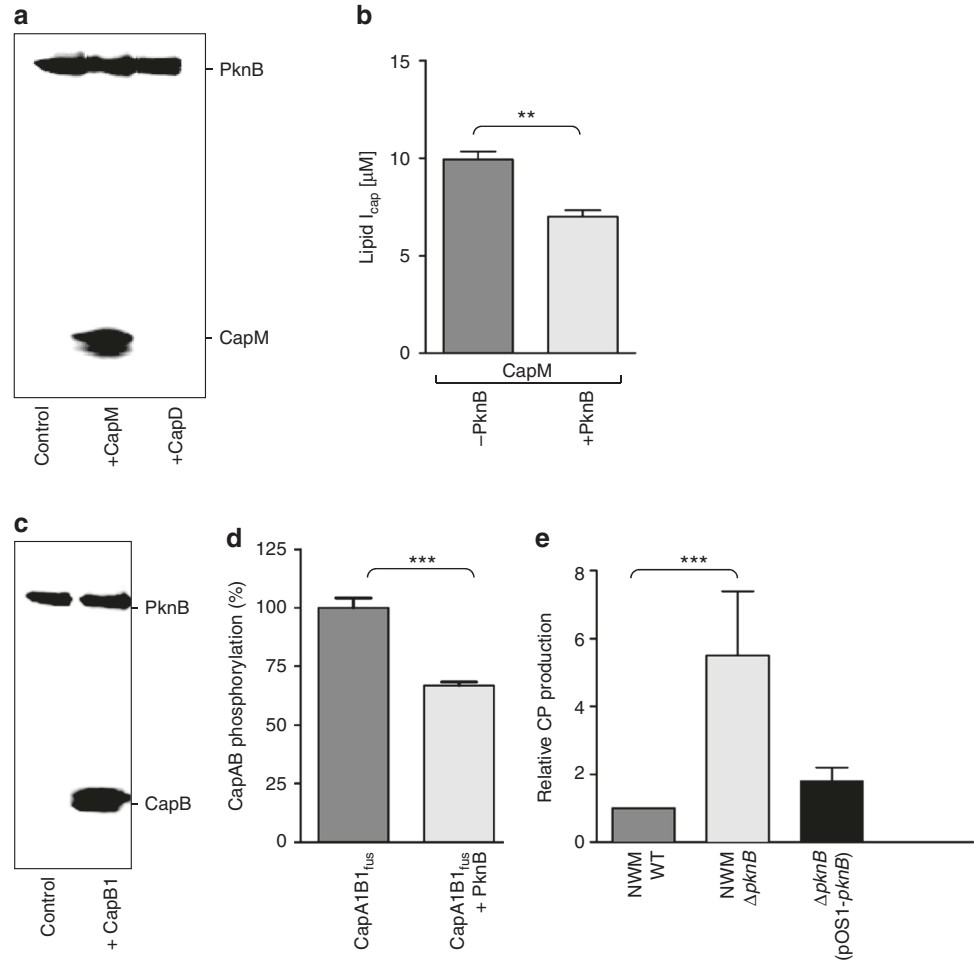

**Fig. 6** CapM and CapB1 are protein targets of the ESTK PknB. **a** CapM is phosphorylated by PknB. Putative target proteins (3 μg) were incubated with PknB in the presence of [$^{33}$P]ATP and reactions were analyzed by SDS-PAGE and phosphoimaging. **b** Impact of PknB kinase activity on CapM GT activity. CapM was incubated in presence of either active PknB (light grey) or heat-inactivated PknB (dark grey) with UDP-D-[$^{14}$C]FucNAc and C$_{55}$P. After extraction with BuOH/PyrAc, reactions were analyzed by TLC and phosphoimaging. Experiments were performed in triplicate. The error bars represent the ± SD from three biological replicates. Statistical significance was analyzed by an unpaired $t$-test (**$p < 0.05$). **c** CapB1 is phosphorylated by PknB. Putative target proteins (3 μg) were incubated with PknB in the presence of [$^{33}$P]ATP, and reactions were analyzed by SDS-PAGE and phosphoimaging. **d** Quantification of CapB autophosphorylation in the absence and presence of PknB. Experiments were performed in triplicate. The error bars represent the ± SD from three biological replicates. Statistical significance was analyzed by an unpaired $t$-test (***$p < 0.005$). **e** ELISA-based quantification of CP production in a Δ$pknB$ deletion strain, the corresponding parental strain $S.$ $aureus$ Newman (WT) and complementation of the Δ$pknB$ mutant $in$ $trans$. Experiments were performed in triplicate. The error bars represent the ± standard error of the mean (SEM) from three biological replicates. Statistical significance was analyzed by an unpaired $t$-test (***$p < 0.005$)

Reversible protein phosphorylation appears an elegant mechanism to ensure a coordinated and temporally controlled flux of intimately shared cell envelope metabolites. Several biosynthetic enzymes involved in polysaccharide production have been identified as endogenous substrates of bacterial tyrosine kinases, for instance the UDP-glucose 6-dehydrogenases TuaA and YwqF of *Bacillus subtilis*[54], the homologous *E. coli* UDP-glucose dehydrogenase Ugd[55] and the UDP-ManNAc dehydrogenase CapO[17] involved in *S. aureus* CP biosynthesis.

The fact that CapO, involved in the third CP reaction cascade (Fig. 1), is a regulatory target of the CapAB kinase complex[17], led us to assume that additional checkpoints within the biosynthetic pathway are also controlled by CapAB, which turned out to be the case for CapE and CapM. Four tyrosine residues in the dehydrogenase CapE were identified as the specific phosphorylation sites, with Tyr76, Tyr290 and Tyr293 being widely conserved among homologous proteins from different pathogens. Interestingly, these residues are located in strategic functional regions of CapE: next to the cofactor binding site, in close proximity to the active site, and within a recently described mobile loop (Fig. 3f). Crystal structure analysis identified this mobile latch to connect two CapE protomers within the hexameric complex (trimer of dimers) and showed that the latch of one dimerization partner is associated with the substrate-binding domain of the contiguous CapE monomer, and *vice versa*, suggesting that this mobile loop is involved in regulating the access of the UDP-D-GlcNAc substrate to the active site[56]. Since contacts to Tyr290 and Tyr293 appear to be involved in the interaction, phosphorylation might induce conformational changes that facilitate access of UDP-D-GlcNAc to the active site, thereby increasing CapE enzymatic activity.

For the monotopic GT CapM, we identified the conserved tyrosine residue at position 157 as the CapA1B1 regulatory phosphorylation site which is in good agreement with the work of Minic et al. (2007) and supports the hypothesis that phosphorylation of the equivalent site in the *S. thermophilus*

phosphogalactosyl-transferase EpsE results in activation in vivo[57]. A recent structural analysis showed, that the equivalent tyrosine residue in PglC is involved in a hydrogen-bonding network that establishes intramolecular interactions between helices A, F and G[37] (Supplementary Figure 3c), suggesting that the phosphorylated amino acid side chain may affect relative strengths of hydrogen bonds and critical interactions in CapM.

Current hypotheses suggest a multifaceted role for BY-kinases in bacterial exopolysaccharide production. BY-kinase signalling may not only regulate the catalytic activity of polysaccharide biosynthetic enzymes, but also ensure the correct cellular localization of protein targets[58]. Moreover, it has been suggested that BY-kinases may interact with the export and polymerization machinery to control the level of CP production and/or CP chain length[52,59]. However, the exact mechanism through which this control is exerted, as well as the molecular signal sensed, remains enigmatic so far.

The phosphodiesterase activity of CapA1 described here is in good agreement with previous studies demonstrating the release of CP and WTA into the culture supernatant in LCP deletion mutants of different species[60–63]. An *lcpC* deletion mutant of *S. aureus* accumulated CP in the culture supernatant, indicating that LcpC is the key LCP enzyme for attachment of CP to the peptidoglycan in *S. aureus*[21].

This study provides biochemical proof that LcpC catalyzes the attachment of CP to PG. Importantly, CP lipid precursor cleavage and transfer of the phosphosugar moiety were found to be enhanced in the presence of CapA1, suggesting that the transmembrane activator cooperates with LcpC by forming an interaction complex, thereby modulating the attachment of CP to the cell wall precursor. In line with this interpretation, Toniolo et al. (2015) reported that the extracellular domain of the CapA1 homolog CpsC of *Streptococcus agalactiae* may modulate LCP-mediated CP attachment in response to the phosphorylation state of the BY kinase CpsD[64].

The natural PG acceptor substrate of LcpC is elusive; possible acceptor structures include the ultimate PG precursor lipid II, as well as "nascent" and crosslinked PG. More recently, LcpA and LcpB were shown to attach a shortened, soluble ($C_{20}P$) WTA precursor to a preformed "nascent PG" oligomer in vitro[65], but not to a lipid $II_{PG}$ mimetic lacking the natural undecaprenyl tail. In contrast, the biochemical analyses presented here clearly demonstrate that CP is efficiently attached to lipid $II_{PG}$ by the LcpC enzyme, and that PBPs are able to polymerize the resulting reaction product. With regard to the membrane localization of both, substrates and enzymes, and the interdependence and intimate connection of the enzymatic machineries, lipid $II_{PG}$ seems the most plausible acceptor.

CP lipid intermediates were further hydrolyzed by LcpC in the absence of the acceptor substrate as predicted by structural analysis of Cps2A of *S. pneumoniae*[20], which was not observed for LcpA and LcpB proteins[65]. In vitro, all three CP lipid intermediates were efficiently processed by LcpC, although the proximal full-length undecaprenyl-pyrophosphoryl-linked sugar moiety appears sufficient for CP precursor recognition. Likewise, the first $C_{55}P$-linked sugar unit of the O-antigen repeat unit contains the recognition information necessary for catalysis by the O-antigen ligase WaaL[66].

The biosyntheses of the cell envelope components PG, WTA and CP have to be coordinated in time and space, since the enzymatic machineries and their individual components are functionally related to each other or intimately connected[1,53]. The complex interplay between different cell envelope pathways becomes evident upon inhibition of individual biosynthetic steps. In analogy to the WTA biosynthesis in *S. aureus*, where late stage biosynthetic genes have been shown to be conditionally essential

("essential gene paradox";[67]), Yother and co-workers reported that deletion of the late stage CP biosynthesis genes, responsible for side chain assembly, polymerization or transport (*cps2K*, *cps2J*, and *cps2H*) in *Streptococcus pneumoniae*, is lethal[68]. Since the capsule is not required for cell viability per se, the damage to the cell envelope is most likely due to an inhibitory effect on PG biosynthesis, resulting from sequestration of CP lipid intermediates probably reducing the undecaprenyl-phosphate level to a critical point where PG synthesis is affected. Since we found CapA1 to hydrolyze CP lipid intermediates even in the absence of LCP proteins and that depletion of CapA1 in a *S. aureus* Δ*lcp* triple mutant was lethal, we conclude that CapA1-mediated cleavage of dead-end lipid-linked products might serve as a rescue mechanism counteracting depletion of $C_{55}P$ to critical levels.

The ESTK PknB is another important player in the orchestration of cell wall polymer biosynthesis and has been implicated in coordinating PG cross-wall formation, autolysis and cell division in *S. aureus*[48–50]. Our findings that PknB downregulates CapM and with the BY-kinase CapA1B1 operating in an antagonal manner underpins the importance of CapM as crucial enzymatic checkpoint. Moreover, in vitro kinase assays indicate that PknB-mediated phosphotransfer onto CapB1 modulates the activity of the CapA1B1 BY-kinase complex itself, thus identifying another activity by which PknB may exert control on CP biosynthesis. Cross-phosphorylation of ESTKs and BY-kinases has also been reported in *B. subtilis*[69]. The CapB homolog PtkA was identified as in vitro phosphorylation target of the ESTK PrkC[69]. Moreover, PtkA autophosphorylation in vivo was strongly enhanced in a Δ*prkC* strain, suggesting that PtkA BY-kinase activity is negatively regulated by PrkC-mediated phosphotransfer[70].

Corroborating the in vitro results, CP production was substantially elevated in a *pknB* deletion mutant of *S. aureus*. Besides decreasing the activity of CP biosynthesis proteins by direct phosphorylation, PknB kinase activity may serve to increase the cellular concentration of UDP-MurNAc-pentapeptide and lipid $II_{PG}$[71], cell wall metabolites that were identified as inhibitors of CapD enzymatic activity[19]. Moreover, PknB signaling was reported to influence the transcription of the *cap5* gene cluster[72], suggesting that this sensor kinase is part of a multilayered signal-transduction network that coordinates CP formation in space and time.

The finding that CP and PG production are inversely controlled by PknB further substantiates a key role for PknB in the coordinate regulation of Gram-positive cell surface glycopolymers. In exponentially growing cells, PknB-mediated Ser/Thr phosphorylation positively controls PG biosynthesis, while exerting negative control on CP production (Fig. 7).

With respect to biosynthesis of the *S. aureus* CP, the CapA1B1 complex may be particularly important for short-term regulation. More recently, Elsholz and co-workers showed that exopolysaccharide (EPS) production in *Bacillus subtilis* is subject to a positive feedback loop that ties the synthesis of the polymer to its own concentration[46]. Regulation of EPS synthesis is mediated by the EpsAB tyrosine kinase complex, whereby direct recognition of EPS by the extracellular domain of the membrane component EpsA seems to result in stimulation of kinase activity[46].

The activator/phosphodiesterase CapA1 directly and specifically interacts with lipid-bound CP precursors, and CapA1B1 signaling stimulates the activity of CP biosynthetic enzymes, triggering enhanced CP production in late-exponential and stationary phase (Fig. 7).

*S. aureus* PknB specifically interacts with lipid $II_{PG}$, likely responding to cellular pool levels of the ultimate peptidoglycan precursor[51]. It is thus conceivable that in order to balance the pool levels of the lipid $II_{PG}$ acceptor and the CP donor substrate,

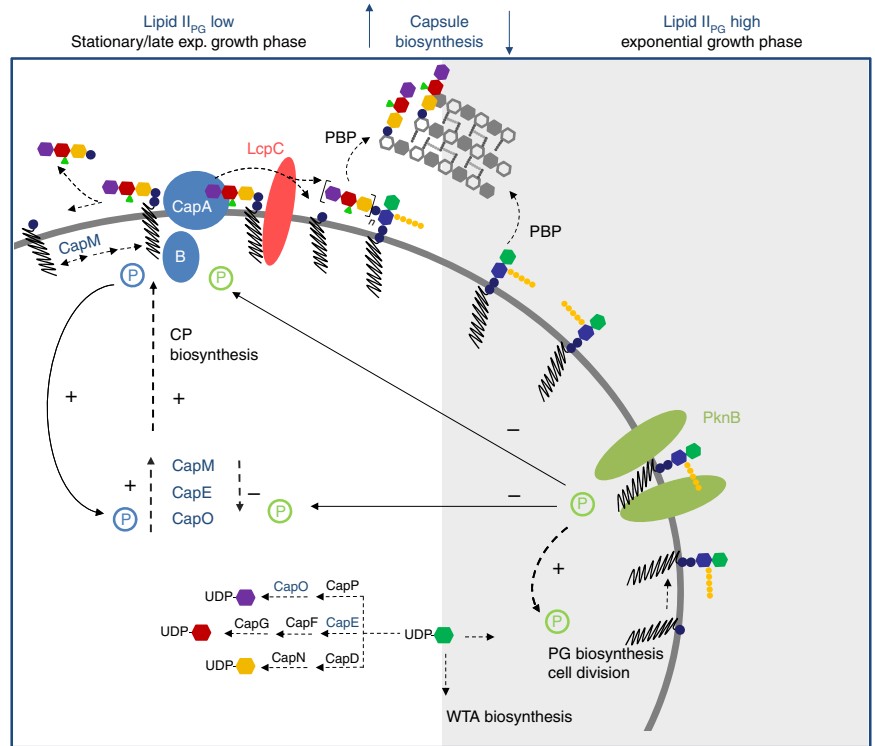

**Fig. 7** Model for the regulation of CP biosynthesis and the attachment to PG in *S. aureus*. The CapA1B1 tyrosine complex (blue) positively controls multiple enzymatic checkpoints to orchestrate CP biosynthesis and precursor consumption during late exponential growth phase (white box). By catalyzing tyrosine phosphorylation (blue ´P´) of biosynthetic enzymes CapE, CapO and CapM, CapA1B1 positively controls the synthesis of both soluble and lipid-linked CP precursors. The membrane-anchored component CapA1 possesses a dual function; CapA1 recognizes lipid-bound CP building blocks and catalyzes the cleavage of the pyrophosphate linkage, releasing the essential isoprenoid lipid carrier $C_{55}P$. CP transfer is catalyzed by LcpC (red), which uses the ultimate peptidoglycan (PG) precursor lipid $II_{PG}$ as an acceptor substrate. During exponential growth phase (grey shading), the insertion of new PG material ($LII_{PG}$) into the cell wall network is increased; UDP-GlcNAc and $C_{55}P$ are preferably channeled into PG and WTA biosynthesis, while CP production is low. The Ser/Thr kinase PknB (green) contributes to the control of cell wall biosynthesis and cell division. Control of CP formation during ongoing PG synthesis is achieved by PknB via Ser/Thr phosphorylation (green ´P´). PknB negatively controls the activity of the priming glycosyltransferase CapM to regulate the initial $C_{55}P$-consuming reaction within the CP biosynthetic pathway. Furthermore, PknB cross-talks with the CapAB complex, thus allowing control of the CP biosynthetic machinery on multiple levels, in response to the status of PG biosynthesis. The model highlights central biosynthetic precursors as important signal molecules situated at the interface of different pathways, integrated in feedback loops to control biosyntheses and the flux of shared precursors

which are both built on the lipid carrier $C_{55}P$, PknB and CapA1B1 antagonistically control biosyntheses to ensure vital cycling of the essential $C_{55}P$ lipid carrier. In addition, CapA1 might contribute to the correct localization of the CP assembly machinery, as proposed for *S. agalactiae*[64], which has to integrate precursor transport and CP polymerization. This interaction protein network thus appears to contribute to the spatiotemporal coordination of CP biosynthesis with PG synthesis (Fig. 7), rendering proteins such as CapA1 and LcpC, exposed at the cell surface and devoid of eukaryotic homologs, potentially attractive antibacterial or antivirulence targets.

## Methods

**Strains**. *S. aureus* Newman NCTC8178 was maintained on blood or lysogeny broth (LB; Oxoid) agar. *Escherichia coli* strains BL21 and C43 used for over-expression of recombinant His6-tag fusion proteins were maintained on LB-agar plates containing 50 μg ml⁻¹ ampicillin or 25 μg ml⁻¹ kanamycin. CP5 production was quantified from *S. aureus* cultivated on Columbia medium supplemented with 2% NaCl and 1.5% agar. The Newman Δ*pknB* mutant was kindly provided by Dr. Knut Olsen (Wuerzburg, Germany), and RN4220 carrying pCapA1 was provided by Dr. Gabriele Bierbaum (Bonn, Germany)[44]. *S. aureus* strain MW2 (NRS123) and strains from the Nebraska transposon mutant library[73] were obtained from the Network on Antimicrobial Resistance in *Staphylococcus aureus* program, which was supported by the National Institute of Allergy and Infectious Diseases of the National Institutes of Health (contract HHSN272200700055C). The Nebraska library is comprised of derivatives of the serotype 5 strain USA300 LAC (cured of

three plasmids) in which individual nonessential genes were disrupted by the insertion of the mariner transposon (Tn) *bursa aurealis* and includes mutants in *capA2* (NE1286), *capB2* (NE75), *capA1* (NE302a), and *capB1* (NE135). Construction of *cap* deletion mutants is described in the Supplementary Methods. The *pknB* and Tn mutants were maintained on tryptic soy agar (TSA) plates with 5 μg ml⁻¹ erythromycin, and pCapA1 was maintained in *S. aureus* cultivated on TSA with 10 μg ml⁻¹ chloramphenicol. *S. aureus* Newman Δ*lcpABC* mutants were kindly provided by Dr. Olaf Schneewind (Chicago, USA)[21] maintained on LB-agar plates. *S. aureus* strains carrying the plasmid pEPSA5 or the antisense plasmid pEPSA5-*capA1AS* were maintained on TSA-agar plates supplemented with 34 μg ml⁻¹ chloramphenicol.

**Plasmid construction and site-directed mutagenesis**. Oligonucleotide primers were purchased from Eurofins MWG Operon (Germany). Primer pairs used for amplification of genomic sequences encoding *capA1*, *capA2*, *capB1*, *capB2*, *capA1B1fus*, *capA2B2fus*, *capA1as*, *capC1*, *capC2*, *capI*, *capL*, *capN*, *capM* and *lcpC* are listed in Supplementary Table 1. PCR reactions were carried out using Phusion DNA polymerase (NEB) and genomic DNA of the serotype 5 strain *S. aureus* Newman as template. PCR products were digested with type II restriction endo-nucleases (NEB; Supplementary Table 1) and ligated (T4 DNA ligase, Roche) with appropriately restricted expression vectors (Novagen). The PCR amplicon of the gene *capN* was inserted into pET24a. The vector pET21b was utilized for cloning of the *capI* and *capL* genes. The antisense fragment of *capA1* was inserted into pEPSA5. All other amplicons were introduced into pET28a. The resulting plasmids (Supplementary Table 2) were confirmed by DNA sequencing (Sequiserve, Germany). Plasmids pET24a-*capD*, pKBK50d, pET5F1.1, pKBK6a, pET28a-*pglF* and pET52b-*pknB* used for overexpression of CapD-His6, CapE-His6, CapF-His6, CapG-His6, PglF-His6 and PknB-His6 have been described previously[13,19]. To

enable expression of N-terminally His$_6$-tagged CapA1B1 and CapA2B2 fusion proteins, the constructs capA1B1fus and capA2B2fus were amplified using the primer pairs capA1_F and capB1_R, and capA2_F and capB2_R, respectively. The QuikChange Lightning Mutagenesis Kit (Stratagene) was used according to the manufacturer's instructions to remove the stop codon of capA by site-directed mutagenesis (TTA > GCA); the respective mutagenesis primers (capA1B1fus_-mut_F, capA1B1fus_mut_R) and capA2B2fus_mut_F and capA2B2fus_mut_R (TAA > TAT) are given in Supplementary Table 1. Site-directed mutagenesis of plasmid pET28a-capM was performed to introduce amino acid exchanges Tyr75 (TAT) > Phe (TTT) (primers: capM_mutY75_F, capM_mutY75_R) and Tyr157 (TAC) > Phe (TTC) (capM_mutY157_F, capM_mutY157_R) in CapM. Likewise, plasmid pKBK50d was mutated to introduce amino acid exchanges Tyr76 (TAC) > Phe (TTC) (capE_mutY76_F, capE_mutY76_R), Tyr290/293 (TAT) > Phe (TTT) (capE_mutY290/293_F, capE_mutY290/293_R), and Tyr329 (TAT) > Phe (TTT) (capE_mutY329_F, capE_mutY329_R) into CapE. Site-directed mutagenesis of plasmid pET28a-capA1B1 was performed to introduce amino acid exchanges Thr8 (ACA) > Glu (GAA) (primers: capA1B1fus_mutT8E_F, capA1B1fus_mutT8E_R) and Thr8 (ACA) > Ala (GCA) (primers: capA1B1fus_mutT8A_F and capA1B1-fus_mutT8A_R) into CapA1B1$_{fus}$. In order to produce a CapB2_T8E mutant, site-directed mutagenesis of plasmid pET28a-capB2 was performed using the primer pair capB2_mutT8E_F and capB2_mutT8E_R to introduce an amino acid exchange Thr8 (ACA) > Glu (GAA). PCR-mediated base pair exchange was confirmed by sequencing. The resulting plasmids, which were utilized for expression and purification of CapM and CapE mutant proteins containing single or multiple Tyr to Phe exchanges, are listed in Supplementary Table 2. pCapA1 was transduced from S. aureus RN4220 to strain MW2 with phage 11, with selection on medium containing 10 µg ml$^{-1}$ chloramphenicol. S. aureus Newman ΔlcpABC was transformed by electroporation[74] with pEPSA5-capA1AS purified from E. coli DC10B, with selection on TSA containing 2.5% glucose and 34 µg ml$^{-1}$ chloramphenicol. Expression of the appropriate antisense fragment was induced on TSA-agar plates containing 500 mM xylose.

**Overexpression and purification of recombinant proteins.** E. coli strain BL21 (DE3) (Promega) was used as host for the recombinant expression of cytoplasmic enzymes (CapB1-His$_6$, CapB2-His$_6$, CapC1-His$_6$, CapC2-His$_6$, CapE-His$_6$, CapF-His$_6$, CapG-His$_6$, CapO-His$_6$,CapP-His$_6$ and PglF-His$_6$) and E. coli strain C43 (DE3) (Promega) was used for the recombinant expression of integral membrane or membrane associated proteins (CapA1-His$_6$, CapA2-His$_6$, CapAB-His$_6$ fusion constructs, CapI-His$_6$, CapL-His$_6$, CapM-His$_6$, CapD-His$_6$, LcpC-His$_6$ and Strep-tagged PknB. E. coli strain BL21 (DE3) cells were grown at 37 °C in lysogeny broth (LB; Oxoid) and E. coli C43 (DE3) cells were grown at 30 °C in double-strength yeast extract-tryptone broth (2YT, Difco) containing the appropriate selective antibiotic (50 µg ml$^{-1}$ ampicillin or 25 µg ml$^{-1}$ kanamycin). At an OD$_{600}$ of 0.6, IPTG was added at a final concentration of 0.5 mM to induce expression of the recombinant protein. Expression times and temperatures were optimized for the individual constructs to ensure high yields of the different fusion proteins. The cultures for overexpression of CapA1, CapA2, CapAB fusion constructs, CapD, CapI, CapL, CapN, LcpC and PknB were induced at 20 °C for 16 h. All other E. coli BL21 (DE3) cultures and the E. coli C43 (DE3) cultures for the expression of CapM were induced at 30 °C for 4 h. After induction, cells were harvested (15 min, 4 °C, 7000 × g) and resuspended in lysis buffer (50 mM Tris-HCl, 300 mM NaCl, pH 7.5), which was supplemented with 1% (v/v) Triton X-100 (Sigma-Aldrich) for purification of CapA1, CapA2, CapAB fusion constructs and CapN or with 29 mM n-dodecyl-β-D-maltoside (DDM, Glycon Biochemicals) for purification of CapD, CapI, CapL, CapM and LcpC. Lysozyme (250 µg ml$^{-1}$), DNase (50 µg ml$^{-1}$) and RNase (10 µg ml$^{-1}$) (Sigma-Aldrich) were added to the suspension; cells were incubated on ice for 30 min and sonicated. Cell debris was removed by centrifugation (20 min, 4 °C, 21,000 × g).

The supernatant was incubated with Ni-NTA-agarose slurry (Qiagen, Germany) for 2 h at 4 °C under gentle stirring. The mixture was then loaded onto a column support. After washing with lysis buffer, weakly bound material was removed with 10 and 20 mM imidazole. Recombinant proteins were eluted with buffer containing 300 mM imidazole. Five 500-µl fractions were collected each and stored in 30% (v/v) glycerol at −20 °C; preparations of CapE were dialyzed against 10 mM potassium phosphate buffer (KPi), pH 7.5 before storage. For PknB purification the pellet was resuspended in lysis buffer containing 1% (v/v) Triton X-100 and the suspension was incubated on ice for 1 h. Cell debris was spun down (20 min, 4 °C, 34,000 x g) and the clear supernatant was loaded onto a Strep-tactin agarose column (IBA, Germany). The column was washed twice with lysis buffer and PknB was eluted with Strep-tactin elution buffer (IBA, Germany). Purity of elution fractions was assessed by SDS-Page (NuPAGE; Invitrogen); protein concentrations were measured using Bradford reagent (Bio-Rad, Germany). Expression and purification of CapA1B1$_{fus}$, CapE and CapM protein variants were carried out as described for the wild-type proteins.

**In vitro syntheses of soluble CP precursors.** UDP-D-GlcNAc and cofactors for enzyme assays were obtained from Sigma-Aldrich. Synthesis of the precursor UDP-2-acetamido-2,6-dideoxy-D-xylo−4-hexulose for mass spectrometric analysis was performed with a recombinant truncated version of the enzyme PglF from C. jejuni, as previously described[19]. The PglF reaction product was used as

substrate for reconstitution of CapN catalytic activity. For this purpose, PglF reactions were carried out overnight and quenched by heating (5 min, 95 °C). CapN catalyzed synthesis of UDP-D-FucNAc was carried out in a total volume of 40 µl. CapN (12 µg) was incubated in the presence of ~3 mM UDP-2-acetamido-2,6-dideoxy-D-xylo−4-hexulose, 1.875 mM NADPH, 8% (v/v) Triton X-100 and 10 mM KPi, pH 7.5, for 2 h at 30 °C. Synthesis of larger quantities of soluble capsule precursors for MALDI-TOF MS analysis was achieved by a 10-fold upscale of the synthesis reaction.

Alternatively, synthesis of UDP-D-FucNAc was performed in an one-pot-assay containing 4 µg CapD, 4 µg CapN, 2 mM UDP-D-GlcNAc, 2 mM NADPH and 0.5 mM target protein in 50 mM Tris-HCl, 10 mM MgCl$_2$, pH 7.5, in a total volume of 30 µl. UDP-L-FucNAc was synthesized by incubating CapE, CapF and CapG in presence of 2 mM UDP-D-GlcNAc, 2 mM NADPH and 0.5 NADP in 50 mM Tris-HCl, 10 mM MgCl$_2$, pH 7.5.

The third soluble precursor UDP-D-ManNAcA was synthesized by incubating 2 mM UDP-D-GlcNAc with CapO and CapP (4 µg each), 6 mM NAD$^+$ and 0.5 mM dithiothreitol (DTT; Sigma-Aldrich) in 50 mM Tris-HCl, 10 mM MgCl$_2$, pH 7.5. All enzymatic synthesis reactions were quenched by heating 5 min, 90 °C).

**In vitro modulation of CapE catalytic activity by CapAB.** The influence of CapAB-mediated tyrosine phosphorylation on CapE catalytic activity was examined in vitro using purified recombinant proteins. Assays were performed in a total volume of 100 µl containing 17 µg of CapE (or of a CapE protein variants) and 8.5 µg of purified CapAB kinase complex. The enzymes were incubated in the presence of 3 mM UDP-D-GlcNAc and 1 mM ATP, in 10 mM KPi, 10 mM MgCl$_2$, pH 7.5 supplemented with 1 mM DTT, for 30 min at 30 °C. Reactions were quenched by heating (5 min, 90 °C), then subjected to capillary electrophoresis (CE) analysis[19]. Control reactions were performed with heat-inactivated (10 min, 100 °C) CapAB protein.

**MALDI-TOF mass spectrometric analysis.** Soluble capsule precursors were purified by RP18-HPLC[19], and cleaved C$_{55}$P intermediates were purified by preparative TLC. Lipid spots were visualized using iodine vapour and extracted off the silica plates with methanol. Samples were spotted onto a ground steel MALDI-TOF target plate and allowed to dry at room temperature. Subsequently, each sample was overlaid with 1 µl of matrix (saturated solution of 6-Aza-2-thiothymine in 50% (v/v) ethanol, 10 mM diammonium hydrogen citrate) and air dried at room temperature again. Spectra were recorded either in the reflector negative mode within a mass range from 300 to 3000 Da (soluble capsule precursors) or in the linear negative mode within a mass range from 400 to 3000 Da (lipid intermediates), at a laser frequency of 9 Hz on a Biflex III mass spectrometer (Bruker Daltonics). Data analysis was performed using flexAnalysis software (Bruker Daltonics).

**In vitro kinase assays.** In vitro BY-kinase assays were carried out in a total volume of 10 µl containing either 0.5 µg of CapA and CapB, or 2 µg of a CapAB protein fusion construct. For identification of protein substrates, 2 µg of a recombinant target protein were added. The proteins were incubated in the presence of 10 µCi γ-labeled [$^{33}$P]ATP (~300 nM; Hartmann Analytic) in 50 mM Tris-HCl, 10 mM MgCl$_2$, pH 7.5 supplemented with 0.5 mM DTT, 0.5 mM EDTA and 10 µM ATP. Assays with CapAB protein variants were carried out as described for the wild-type proteins.

For identification of PknB protein substrates, purified PknB (0.5 µg) was incubated in presence of 3 mM MnCl$_2$ in the reaction mixture described above. Cross-phosphorylation of CapAB by PknB was assessed analogously.

After 30 min of incubation at 30 °C, reactions were stopped by addition of 4x LDS sample buffer (Invitrogen), and analyzed by SDS-PAGE (NuPAGE, Invitrogen). Radioactive protein bands were visualized using a storage phosphor screen in a Storm imaging system (GE Healthcare). A detailed description of the identification of phosphorylation sites by nanoLC-MS/MS is given in the Supplementary Methods.

**In vitro phosphatase assays.** CapC phosphatase activity was examined in vitro by the addition of CapC1 or CapC2 (2 µg each) with 2 mM MnCl$_2$ to the CapAB in vitro kinase assay (see above) and subsequent heat-inactivation. After incubation for 1 h at 30 °C, phosphatase reaction was stopped by the by addition of 4x LDS sample buffer (Invitrogen), and analyzed by SDS-PAGE (NuPAGE, Invitrogen). Radioactive protein bands were visualized using a storage phosphor screen in a Storm imaging system (GE Healthcare).

**In vitro syntheses of lipid I$_{cap}$, lipid II$_{cap}$, and lipid III$_{cap}$.** Lipid I$_{cap}$ synthesis was carried out in a total volume of 50 µl containing 5 nmol C$_{55}$P (Larodan, Sweden), 2 mM UDP-D-GlcNAc, 10 mM NADPH, 0.5 mM NADP and 0.6% (v/v) DMSO in 50 mM Tris-HCl, 10 mM MgCl$_2$, pH 7.5. The reaction was initiated by the addition of 4 µg of biosynthetic enzymes CapD, CapN and CapM, and incubated for 16 h at 30 °C.

To assess the influence of different detergents and phospholipids on CapM activity, newly synthesized UDP-D-FucNAc (described above) was incubated with 4 µg CapM, 5 nmol C$_{55}$P in a reaction mixture containing 50 mM Tris-HCl, 10 mM

$MgCl_2$, pH 7.5 supplemented with an appropriate amount of triton X-100, n-dodecyl-β-D-maltoside (DDM), n-lauroyl sarcosine (LS), dimethylsulfoxide (DMSO), dioleoylphosphatidylglycerol (DOPG) or cardiolipin (CL). To investigate the modulatory effect of CapAB on CapM activity 4 μg CapAB and 10 mM ATP were added to the CapM reaction mixture. To assess the influence of PknB on CapM activity in vitro, 1 μg PknB was added to the CapM reaction mixture supplemented with 2 mM $MnCl_2$ and 10 mM ATP.

The second precursor lipid $II_{cap}$ was synthesized in a total volume of 50 μl by incubating the capsule biosynthesis proteins CapE, CapF, CapG and CapL (4 μg each) in the presence of 2 nmol purified lipid $I_{cap}$ and 2 mM UDP-D-GlcNAc, 2 mM NADPH, 0.5 mM NADP and 0.6% (v/v) DMSO in 50 mM Tris-HCl, 10 mM $MgCl_2$, pH 7.5.

Lipid $III_{cap}$ was synthesized by incubating 2 nmol purified lipid $II_{cap}$ with CapO, CapP and CapI (4 μg each) and 2 mM UDP-D-GlcNAc, 6 mM $NAD^+$, 0.5 mM dithiothreitol (DTT; Sigma-Aldrich) in 50 mM Tris-HCl, 10 mM $MgCl_2$, pH 7.5. Synthesized CP precursors were extracted from the reaction mixture with an equal volume of n-butanol/pyridine acetate, pH 4.2 (2:1, v/v) and analyzed by TLC on silica plates (Merck) according to Rick (chloroform, methanol, water, ammonium hydroxide, 88:48:10:1)[75]. Nisin was added at a molar ratio of 2:1 with respect to the lipid precursor prior to the extraction procedure. Reaction mixtures were extracted and unbound lipids were analyzed by TLC. For mass spectrometric analysis of lipid intermediates see Supplementary Methods.

For synthesis of [$^{14}$C]-labeled lipid-bound precursors, the assays were further supplemented with 0.333 nmol UDP-D-[$^{14}$C]GlcNAc (Hartmann Analytic, Germany). Radiolabeled spots were visualized using a storage phosphor screen in a Storm imaging system (GE Healthcare). Non-radiolabeled lipid intermediates were analyzed using PMA staining reagent (2.5% (w/v) phosphomolybdate, 1% (w/v) ceric sulfate, 6% (v/v) sulfuric acid). Isolation of small quantities of CP precursors was achieved by synthesis and subsequent purification via preparative TLC. To this end, lipid spots were visualized using iodine vapour and material was scratched off the silica plates. Lipids were extracted by incubation in 100 μl of methanol for 60 min. Larger quantities of CP precursor were purified using high-performance liquid chromatography (HPLC) over a DEAE-FF (5 ml; GE Healthcare) and eluted in a linear gradient from chloroform/ methanol/ water (2:3:1) to chloroform, methanol, 300 mM ammonium bicarbonate (2:3:1).

**CapA1 catalyzed cleavage of lipid-bound CP precursor**. For reconstitution of the CapA1 mediated cleavage of the pyrophosphate-linkage of lipid-linked cell wall intermediates, purified CapA1 (8 μg) was incubated with 2 nmol lipid $I_{cap}$, lipid $III_{WTA}$, lipid $II_{PG}$ in 10 mM $MgCl_2$ and 50 mM Tris-HCl, pH 7.5. After incubation for 16 h at 30 °C, cleavage products were extracted from the reaction mixture with an equal volume of n-butanol/pyridine acetate, pH 4.2 (2:1, v/v) and analyzed by TLC on silica plates (Merck) according to Rick (chloroform, methanol, water, ammonium hydroxide, 88:48:10:1) and visualized by PMA staining[75].

**In vitro LCP assays**. For reconstitution of the CP ligation reaction purified lipid $I_{cap}$ or [$^{14}$C]lipid $I_{cap}$ was incubated in the presence of LcpC (4 μg), using 2 nmol of purified lipid $II_{PG}$ or [$^{14}$C]lipid $II_{PG}$ as acceptor substrate in a 50 μl reaction mixture with 0.6% DMSO, 18 μg DOPG, 10 mM $MgCl_2$ in 50 mM MES buffer, pH 5.5. Synthesis and purification of the acceptor substrate lipid II is described in the supplementary methods. If indicated, assays additionally contained 4 μg of CapA1. After incubation for 16 h at 30 °C, samples were analyzed by TLC, followed by phosphoimaging. Polymerization of LcpC reaction products were catalyzed by the subsequent addition of PBP2 with 2 mM $CaCl_2$ to the LCP reaction mixture, followed by incubation at 30 °C for 1 h. Moenomycin (MOE) was used to block the PBP2 catalyzed reaction and added at a final concentration to 10 μM. For evaluation of LcpC mediated hydrolysis of CP precursor in the absence of the acceptor substrate, 2 nmol of the individual lipid-linked precursor was incubated with 4 μg LcpC, 18 μg DOPG, 10 mM $MgCl_2$ and 0.6% DMSO in 50 mM MES buffer, pH 5.5. After incubation for 16 h at 30 °C, samples were analyzed by TLC and visualized by PMA staining.

**Reporting summary**. Further information on experimental design is available in the Nature Research Reporting Summary linked to this article.

## Data availability
The authors declare that all data supporting the findings of this study are available within the Article and its Supplementary Information.

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

## Acknowledgements

We would like to thank Michaele Josten for mass spectrometry, and Katie O´Riordan for providing plasmids pKBK4, pKBK6a, pKBK10, pKBK50d, pET24-capD, pET24-capN,

and pET5F1.1. We thank Jerry Pier for providing purified PNAG, Knut Ohlsen for providing strain *S. aureus* Newman Δ*pknB*, Gabriele Bierbaum for providing plasmid pCapA1, Andreas Peschel for providing *E. coli* DC10β and Olaf Schneewind for kindly providing the Δ*lcpABC* mutant. Marvin Rausch is grateful for support by the Juergen Manchot Foundation. This work was supported by the German Research Foundation (DFG; SCHN 1284/1–2) and the BONFOR program of the Medical Faculty of the University of Bonn.

## Author contributions

M.R., J.P.D., H.U., A.M., W.L., P.H., M.S., M.E., X.W., X.L. performed experiments. C.E.M., W.V., H.G.S., J.C.L., T.S. contributed to the study design, planning of the experiments and interpretation of data. H.U. and T.S. wrote the manuscript. M.R., A.M., C.E.M., W.V., J.C.L., H.G.S., T.S. contributed to revision of the manuscript.

## Additional information

**Competing interests:** The authors declare no competing interests.

