## [Peer Review File · Nature Communications]

Reviewers' comments:

Reviewer #1 (Remarks to the Author):

The manuscript "Integration of Capsule Assembly Modules into Cell Wall Biosynthetic and Regulatory Networks" attempts to reconstitute key components of *S. aureus* CP5 biosynthesis in vitro - including initial intermediate synthesis and ultimate transfer to PG- and builds a case that bacterial tyrosine kinases are master coordinators of cell wall biosynthesis through reversible protein phosphorylation. While the research topics covered in this work are truly exciting and of high importance, this reviewer feels the authors have not provided enough evidence to support their conclusions.

Major issues:

1. Several of molecules generated from enzymatic synthesis lack proper characterization. These include the Lipid Icap molecule and the product from LcpC-mediated reaction. Providing at minimum MS analysis (and ideally MS/MS characterization) of these molecules is critical to confirm function of both CapL and LcpC. Elucidating the molecular details of attachment of glycopolymers to PG is of high interest, however, the Rf value of the product generated by LcpC - one member of a family of enzymes thought to facilitate this reaction - in the TLC assays described in this work is higher than Lipid IIPG. If the product of this reaction was truly the appendage of FucNAc-P to Lipid IIPG, as claimed by the authors, then the Rf value of this molecule would be significantly lower than the Lipid IIPG starting material (due to an increase in net negative charge and contributions from the additional hydroxyl moieties). Additionally, there is an obvious signal corresponding to this molecule in the negative control in the second panel of Figure 6. Therefore, without adequate MS data, questions are raised about the identity of this molecule. This reviewer is also sceptical of the reported phosphodiesterase activity of LcpC on LipidIcap. Lcp enzymes are known to purify with polyisoprenoid phosphates when overexpressed in an *E. coli* host and thus could be the source of the signal shown in SFig. 7B.

2. The author's work on BY-kinases CapAB is a strength of the manuscript. As the authors state "reversible protein phosphorylation appears an elegant mechanism to ensure a coordinated and temporally controlled flux of intimately shared cell envelope metabolites". This idea is gaining traction in the literature and the authors provide in vitro data supporting the hypothesis showing that, at least in vitro, phosphorylation of CP5 biosynthetic enzymes CapM and CapE by CapAB kinases has influence on enzymatic activity. The authors further demonstrate that the loop regions of CapA1 and CapA2 sense different signals (whereby CapA2 seems to be affected by polysaccharides), and provide evidence suggesting that CapA1 has specific phosphodiesterase activity towards undecaprenyl-linked cell wall intermediates. Despite well-executed in vitro data, this reviewer feels that there is not enough in vivo data to advance the readers understanding of how these proteins may control the flux of critical precursors during cell wall biosynthesis. Indeed, the author's show that *S. aureus* strains compromised in CapAB function lack cell wall-anchored CP. However, this reviewer was left wondering what other phenotypes these strains display. If CapAB kinases - at least in part - control the flux of precursors and are responsible for the recycling of the critical molecule C55P, then one might expect these compromised strains to: exhibit sensitivity to beta-lactams, have altered cellular pools of PG/WTA precursors, and potentially display cell-shape/division defects. In addition, if the CapA1 phosphodiesterase activity truly does alleviate lethality caused from sequestration of C55P from accumulated CP/WTA precursors seen in a Δ LcpABC background, then a Δ capA1 Δ LcpABC should be non-viable. The authors also note that in vivo work has been conducted (Minic et al. 2007) showing increase in activity of EpsE by BY-kinases. Could similar experiments be conducted for CapM and CapE to support the in vitro data?

Other issues:

3. Throughout the manuscript there are numerous references to fluctuations in enzyme activity; however, there are no quantitative numbers given. The author is left to infer this based largely TLC read-outs. Could Kcat/turnover number be provided? Are these numbers representative to

- enzymes of similar function? Were these experiments conducted in duplicate, triplicate?
4. The reviewer is confused as to what the actual WT levels of CapM-mediated Llcap synthesis are. In SF3 A 20 uM of Llcap are produced, yet in Figure 3 C 5 uM are produced. Were the results in SF3 conducted in the presence of CapAB? If so, all the experiments conducted in panel C could have been affecting the phosphorylation ability of CapAB and not the actual activity of CapM. This needs to be addressed and clarified in text.
 5. P6L16: here show re-word to "we show were that"
 6. MS on SF2 requires better annotation to make results clear. What ion is the 590.497 m/z? [M-H⁺]⁻ or [M+H⁺]⁺? What are the 588.451 and 606.48 m/z species?
 7. F2: A true negative control would show the lack of radioactive species in the lane from no enzyme on the same phosphor screen. Showing the iodine stain is however a nice lane to show the Rf of C55P . This reviewer would be very curious to see where UDP-D-[14C]FucNAc runs in this system. What are the additional signals seen in the CapL lane?
 8. F2B: Results with be strengthened by high-res MS that has a mass accuracy within 0.5 ppm of the calculated mass. In addition MS/MS will help add evidence to the product/structure being proposed. The structure of the polyisoprenoid tail in this structure is drawn incorrectly. Undecaprenyl has 7Z (Cis)-isoprene units and 3E (trans)-Isoprenene units with a terminal unit (=11). The structure drawn here has the wrong configuration and contains an extra CH₃ appendage in the Z-units.
 9. P8L14-15: There is no F.2C in the main text.
 10. F4: The reviewer suggests that Cap5A be labeled as CapA1 as this is what the protein is referred to as in text. This would improve the readability of the manuscript.
 11. CapA1 and CapB2 are paralogs of each other. How similar in amino acid identity are the extended loops? A discussion should be presented as to potential reasons for differences in signal sensing and activities of this region at the molecular level.
 12. Fig1. Attachment of the CP polymer to PG is missing a phosphate molecule in the peptidoglycan meshwork.

Reviewer #2 (Remarks to the Author):

This is an extensive and comprehensive study on capsule polysaccharide (CP) biosynthesis in *S. aureus*. The authors do not only succeed in reconstituting the major steps of this complex pathway in vitro, they also describe how the activity of several of the enzymes is controlled by different protein tyrosine kinases. The manuscript sheds new light on CP synthesis in *Staphylococcus aureus* by providing in depth biochemical characterization of key factors of the molecular machinery. Furthermore it underlines the biochemical findings by using in vivo data derived from analysis of capsule mutants. Especially interesting is the new finding that CapA seems to carry out an additional control function via a phosphodiesterase activity. This study is an important contribution to our understanding of capsule polysaccharide, not only for *S. aureus*, but many bacterial pathogens using related pathways. However, the following points could be improved in the revised manuscript:

Introduction: It is debated in the scientific community whether CP is a major virulence factor in *S. aureus* since major highly virulent strains (CA strains), such as MW2 and USA300, are deficient in CP. This issue could be mentioned and discussed better.

CapC (CapC1 and CapC2) is believed to be a phosphatase, modulating the activity of the transmembrane adaptor CapA and BY-Kinase capB. Please explain why this protein was not included in the experiments and discuss its putative role.

Page 7, Line 12: The mass in Supplementary Figure 2 is indicated as 590.4. However in the text it

is discussed as 591.4. Which is the correct mass? (UDP-FucNAc should have 591.08 according to this web tool <http://www.sisweb.com/referenc/tools/exactmass.htm>)

Line 16-18: Please explain why the negative control was not performed with boiled CapM or CapL?

Line 25-26: Why is the mass for the enzymatic product of CapM provided but not the mass of the following reaction by CapL?

Page 8, Line 24- Page 9 line 1: The authors should mention that tunicamycin inhibits CP synthesis in vivo as well.

Page 9, Line 22 - Page 10 line 4: which CapA and B homologue is "CapAB"? From the figure legend it can be concluded that CapA1B1-fusion was used. This should be clarified in the text.

Line 22-25: It is not clear from the text, why the authors tested only proteins CapD CapN CapM and CapE for phosphorylation by CapAB. CapF and CapG are discussed to be no target of CapAB-phosphorylation but this does not seem to be shown. Please clarify.

Page 10, Line 21-25: The authors describe the structural features of CapE in detail - this could be made more clear by including it in Figure 3.

Page 11, Line 12 -Page 12 Line 4: The text refers to the enzyme as CapA1- however in Figure 4 A+B it is indicated as Cap5A . Please clarify.

Page 12, Line 9-10: Complementation of mutant CapA1 and CapB1 could be included in Figure 5C

Page 14, Line 1-2: It is not clear from the figure (Fig 6 A) why PBP2 is catalytically active.

Page 15, Line 5-7, Figure 7C: The complemented PknB mutant could be included

Discussion: The first paragraphs repeat data and conclusions from the results section and could be considerably shortened.

Page 17, Line 20-24: How can an enhancing effect of CapA1 on LcpC be explained on the molecular level?

Page 18, Line 6: It is not clear from the data why PBP2 is active

Figure 1B: What does the small arrow or double arrow in the cartoon enzymes mean? Please describe in the Figure legend

Figure 3B: Why are there two bands for CapM and CapE (Protein degradation products? Issue is not addressed.)

Figure 4 A+B: Cap5A should be called CapA1 as in the corresponding text

Figure 6: What do we see for PBP2?

Reviewer #3 (Remarks to the Author):

Rausch and colleagues use a biochemical approach to study capsular biosynthesis of

Staphylococcus aureus. Using purified Cap proteins, sugar nucleotide precursors and undecaprenyl-phosphate, the authors demonstrate CapM catalyzes the synthesis of lipid I/cap (C55-PP-FucNAc), that CapAB phosphorylate CapM and CapE to control capsular synthesis, that CapA exerts phosphodiesterase activity on lipid I/cap, that LcpC links lipid I/cap to lipid II/peptidoglycan and that PBP2 polymerizes not only lipid II/peptidoglycan but also lipid II/peptidoglycan/FucNAc, and that the threonine/serine kinase PknB acts on CapM to impact capsular polysaccharide synthesis.

General comments

The paper is well written, interesting to a large audience, and provides new biochemical insights on capsular polysaccharide synthesis. This said, the paper seems also odd in that it describes the CapM initiating reaction which leads to lipid I/cap but not the remainder of the biosynthetic pathway. To this, the authors add the study of CapAB-mediated regulation of CP synthesis, which involves CapM and CapE. The studies on LcpC are even further removed, especially as LcpC is thought to use lipid III/cap as a substrate, not lipid I/cap. With this in mind, the question remains whether LcpC actually catalyzes the addition of lipid III/cap to lipid II or to peptidoglycan. I am not convinced that the observed LcpC synthesis of lipid I/cap with lipid II/PG is meaningful.

Specific comments

1. The Abstract does not capture the experimental work described in this paper.
2. What is the source of C55-P and what is its purity?
3. The Discussion is excessively long, boring and at times redundant with the Results section.
4. If the authors want to keep the LcpC data in the paper, I would advise that the shortcomings of the experimental design are explained in detail.
5. The product of the LcpC catalyzed reaction, C55-PP-PG-FucNAc was not characterized by mass spectrometry. How can the authors be sure the TLC signal represents this product?

Reviewer #4 (Remarks to the Author):

In this manuscript, authors have performed excellent biochemistry to reconstitute capsule biosynthesis *in vitro*. In the process they established biochemical assay for CapM enzymes, paving the way for investigating the regulation of its activity. To investigate the regulation of CP synthesis, authors focused on bacterial tyrosine kinases and the eukaryotic serine/threonine kinase PknB. Authors show that CapB1 is a functional kinase and identify CapM and CapE to be direct targets of CapAB chimera. Further work elucidated CapA1 to be a dual-functional protein with phosphodiesterase and kinase activator activity. Interestingly, despite functional redundancy *in vitro*, CapA2 and CapB2 do not play any role in modulating CP production, indicating specific CP independent function for CapB2 kinase *in vivo*. Overall I believe the manuscript has some very interesting and novel data. *In vitro* experiments are very well performed. I find the regulation mediated through phosphorylation on Tyr residues and Thr residues quite intriguing. I also find the cross-phosphorylation of BY kinase by ESTK quite interesting. However these conclusion are solely based on *in vitro* experiments. I believe that *in vivo* evidence is critical for demonstrating phosphorylation-mediated regulation and the role of cross-phosphorylation in modulating Capsule assembly. My specific comments are appended below.

Major Comments:

It is not clear to the reviewer as to how authors have arrived at Tyr157 as the primary phosphorylation site. There is no mass spec data presented either in the main text or in the supplementary to support this conclusion. This data requires support from mass spec data and *in vitro* kinase assays with the wild type and mutant proteins.

Based on the in vitro experiments authors suggest that Tyr157 on CapM and four tyrosine residues on CapE are target phosphorylation sites. Manuscript lacks in vivo evidence for phosphorylation of substrates (CapM by PknB and CapB; CapE by CapB and CapAB by PknB). It is necessary to show that CapM, CapE and CapB1 (on Thr residues) are indeed phosphorylated in vivo. It is also important to identify the in vivo phosphorylation sites with the help of mass spec and determine the stoichiometry of phosphorylation at different locations (different Thr and Tyr residues).

It is not clear to the reviewer how authors have performed the experiment presented in Fig 7E. I did not see any experiment showing phosphorylation of CapAB by PknB, nor did I see any description of the experiment either in methods or in the legend. How did they differentiate between PknB mediated phosphorylation of CapAB and autophosphorylation of CapAB? Why the experiment was not performed with phospho-ablative and phospho-mimetic mutants to eliminate the ambiguity?

Generation of deltaCapM mutant and complementation experiments with the wild type and mutant and their relative impact on CP production would be necessary to investigate regulatory aspects. Similarly complementation of deltaCapB1 mutant with the wild type CapB1 and PknB target site mutants would be necessary to show a role for cross-phosphorylation in CP production.

Minor Comments:

There is certain amount of redundancy between results and discussion part. Specific example would be Page 10 (lines 19-26) and page 11 (lines 1-2). Almost the same contents are rewritten in the Page 16 (lines 7 to 17). Authors need to rework such parts to make the manuscript more concise.

Loading controls are required for both Figure 3A and 3B. Autophosphorylation and trans-phosphorylation of CapM and CapE bands should be indicated for clarity. In the text authors state that CapD, CapN, CapG and CapF are not phosphorylated. However in the figure authors have neither performed the reaction with CapG nor CapF.

Figure 7D requires CapB1 and CapB2 alone controls.

In Figure 4A, authors label CapA1 as Cap5A. Though elsewhere in the manuscript, authors do state that Cap5a and CapA1 are the same, for the ease of understanding labeling should be kept consistent.

Response to referees

Reviewer #1 (Remarks to the Author):

The manuscript "Integration of Capsule Assembly Modules into Cell Wall Biosynthetic and Regulatory Networks" attempts to reconstitute key components of *S. aureus* CP5 biosynthesis in vitro - including initial intermediate synthesis and ultimate transfer to PG- and builds a case that bacterial tyrosine kinases are master coordinators of cell wall biosynthesis through reversible protein phosphorylation. While the research topics covered in this work are truly exciting and of high importance, this reviewer feels the authors have not provided enough evidence to support their conclusions.

Major issues:

1. Several of molecules generated from enzymatic synthesis lack proper characterization. These include the Lipid II_{cap} molecule and the product from LcpC-mediated reaction. Providing at minimum MS analysis (and ideally MS/MS characterization) of these molecules is critical to confirm function of both CapL and LcpC. Elucidating the molecular details of attachment of glycopolymers to PG is of high interest, however, the R_f value of the product generated by LcpC - one member of a family of enzymes thought to facilitate this reaction - in the TLC assays described in this work is higher than Lipid II_{PG}. If the product of this reaction was truly the appendage of FucNAc-P to Lipid II_{PG}, as claimed by the authors, then the R_f value of this molecule would be significantly lower than the Lipid II_{PG} starting material (due to an increase in net negative charge and contributions from the additional hydroxyl moieties). Additionally, there is an obvious signal corresponding to this molecule in the negative control in the second panel of Figure 6. Therefore, without adequate MS data, questions are raised about the identity of this molecule.

Thank you for the careful review and comments. The reviewer is correct with regard to the migration behavior of the lipid II_{PG}-CP product. We have repeated the assay and found that in line with the increase in negative charge and the addition of hydroxyl moieties the product migrates lower compared to lipid II_{PG}. We included a new figure (see updated Figure 5B). We can only speculate that the composition of the TLC mobile phase was composed incorrect in previous analyses. We further provide MS data verifying the lipid II_{cap} molecule. In addition, we now have reconstituted the entire biosynthesis of CP lipid intermediates and included the synthesis of lipid III_{cap}.

However, despite extensive effort we were unable to obtain MS data for lipid III_{cap} and the full-length LcpC reaction product, due to the fact that both ionize poorly and do not fly. Furthermore, the synthesis of these lipids is not trivial and extremely challenging, limiting the amount of precursor available. We now mention this in the text (page 8, lane 8-9).

Recently the group of Suzanne Walker (Schaefer *et al.*, Nature Chem Biol, 2017) used shortened WTA precursors (with a C20 tail instead of C55) to study soluble versions of LcpA and LcpB enzymes (lacking TM domains) to circumvent some of these issues. However, these shortened, soluble substrate variants are far from being natural substrates, particularly for membrane bound enzymes that use lipid-coupled substrates, posing the risk that enzymatic activities or specificities are misinterpreted or not detected.

Although, we do not present MS data for these products we now provide additional evidence for their identity: (i) in the new TLC (Fig. 2 A) we show that all three CP intermediates migrate with their expected R_f values and (ii) we further verify their identity by showing that they form extraction-stable complexes with the lantibiotic nisin. Nisin binds to the C55-PP-sugar moiety of analogous peptidoglycan and WTA precursors (Müller *et al.*, Microb Drug Resist, 2012).

In addition, we now shown that LcpC efficiently ligates these lipid intermediates, including the ultimate CP precursor lipid III_{cap}, to lipid II_{PG} (see new Figure 5D). Most importantly, and in contrast to previous assumptions (Schaefer *et al.*, 2017), PBP2 is indeed able to polymerize the LCP reaction product and PBP2 catalyzed transglycosylation is inhibited by moenomycin used as a control (Fig. 5).

Moreover, in line with the R_f values, a radiolabeled sugar moiety (ManNAcA) is added to the purified lipid II_{cap} (verified by MS), yielding radiolabeled lipid III_{cap} product.

This reviewer is also sceptical of the reported phosphodiesterase activity of LcpC on Lipid_{cap}. Lcp enzymes are known to purify with polyisoprenoid phosphates when overexpressed in an *E. coli* host and thus could be the source of the signal shown in SFig. 7B.

In the control reaction we used heat-inactivated LcpC (now Suppl. Fig 10, lane 2; see presence of glycerol). The respective band that results from Lipid_{cap} cleavage is missing here and we can thus exclude lipid co-purification. We have specified this in the new figure (Suppl. Fig. 10 C) and figure legend.

2. The author's work on BY-kinases CapAB is a strength of the manuscript. As the authors state "reversible protein phosphorylation appears an elegant mechanism to ensure a coordinated and temporally controlled flux of intimately shared cell envelope metabolites". This idea is gaining traction in the literature and the authors provide *in vitro* data supporting the hypothesis showing that, at least *in vitro*, phosphorylation of CP5 biosynthetic enzymes CapM and CapE by CapAB kinases has influence on enzymatic activity. The authors further demonstrate that the loop regions of CapA1 and CapA2 sense different signals (whereby CapA2 seems to be affected by polysaccharides), and provide evidence suggesting that CapA1 has specific phosphodiesterase activity towards undecaprenyl-linked cell wall intermediates. Despite well-executed *in vitro* data, this reviewer feels that there is not enough *in vivo* data to advance the readers understanding of how these proteins may control the flux of critical precursors during cell wall biosynthesis. Indeed, the author's show that *S. aureus* strains compromised in CapAB function lack cell wall-anchored CP. However, this reviewer was left wondering what other phenotypes these strains display. If CapAB kinases - at least in part - control the flux of precursors and are responsible for the recycling of the critical molecule C55P, then one might expect these compromised strains to: exhibit sensitivity to beta-lactams, have altered cellular pools of PG/WTA precursors, and potentially display cell-shape/division defects. In addition, if the CapA1 phosphodiesterase activity truly does alleviate lethality caused from sequestration of C55P from accumulated CP/WTA precursors seen in a Δ LcpABC background, then a Δ capA1 Δ LcpABC should be non-viable. The authors also note that *in vivo* work has been conducted (Minic et al. 2007) showing increase in activity of EpsE by BY-kinases. Could similar experiments be conducted for CapM and CapE to support the *in vitro* data?

We appreciate the reviewers comment and positive feedback. To strengthen the role of CapA1 in the cellular context we now show that depletion of CapA1 in a triple Δ lcp *S. aureus* mutant is lethal (new Suppl. Figure 10D). In this context, we now also show that CapA1 is able to hydrolyze the WTA precursor lipid III (new Suppl. Fig 8B), further supporting the proposed cellular role of CapA1. In addition to this newly discovered phosphodiesterase activity, we provided *in vivo* evidence that *S. aureus* lacking CapA1 does not produce CP *in vivo*.

Because of the highly complex regulatory network underlying cell wall biosynthesis, we focused on the individual reactions *in vitro* which involve multiple integral membrane proteins and membrane anchored substrates. We believe that our approach is a unique strength of this manuscript. To address the last point and add *in vivo* data, we set out to generate a CapM mutant in *S. aureus*, which for unknown reasons failed. It might be suggested that CapM has additional functions within CP biosynthesis that go beyond its GT activity. This is currently under investigation and will be the subject of future studies. Nevertheless, we now support our findings on CapM using structural modeling (compare Suppl Fig. 3) based on structural analysis recently published for the homologous glycosyltransferase PglC by the Imperiali group (Ray et al., Nature Chem Biol, 2018).

Other issues:

3. Throughout the manuscript there are numerous references to fluctuations in enzyme activity; however, there are no quantitative numbers given. The author is left to infer this based largely TLC

read-outs. Could Kcat/turnover number be provided? Are these numbers representative to enzymes of similar function? Were these experiments conducted in duplicate, triplicate?

CapE biochemical analysis and kinetics have been published previously by our group, which is cited in the text. For CapM we now determined K_m and V_{max} , in the presence and absence of CapAB (Suppl Figure 6B). To further support the role of CapAB mediated phosphorylation on CapM we also determined K_m and V_{max} for a CapM phosphomimetic in which Tyr157 was exchanged to Glu. Moreover, we determined the IC_{50} value for tunicamycin, which is in the range of other monotypic GTs, such as PglC of *C. jejuni*. The text was changed accordingly on page 11, lines 7-14 and page 9, lines 7-16.

4. The reviewer is confused as to what the actual WT levels of CapM-mediated Llcap synthesis are. In SF3 A 20 μ M of Llcap are produced, yet in Figure 3 C 5 μ M are produced. Were the results in SF3 conducted in the presence of CapAB? If so, all the experiments conducted in panel C could have been affecting the phosphorylation ability of CapAB and not the actual activity of CapM. This needs to be addressed and clarified in text.

Experiments in SF3 (now SF 4) were conducted in the absence of CapAB and in the presence of DMSO. We have clarified this in the figure legend mentioning "CapM *in vitro* activity was tested in the absence of CapAB and in the presence of DMSO." (page 45, lines 20-21).

5. P6L16: here show re-word to "we show were that"

Thank you. Fixed.

6. MS on SF2 requires better annotation to make results clear. What ion is the 590.497 m/z? $[M-H]^-$ or $[M+H]^+$? What are the 588.451 and 606.48 m/z species?

Thank you. We apologize for being imprecise. Annotation has been improved. The text now states (page 7, lines 13-15) "...sugar nucleotide species having a molecular mass of m/z 590.4 for the negatively charged molecule, consistent with the formation of UDP-D-FucNAc." and the figure legend (SF 2) has been extended accordingly (page 45, lines 1-6): "Peaks at m/z 588.4 ($[M-H]^-$) correspond to the deprotonated molecular ion of UDP-2-acetamido-2,6-dideoxy-D-xylo-4-hexulose (neutral mass 589.3). The peak at m/z 590.4 ($[M-H]^-$) corresponds to UDP-D-FucNAc (neutral mass 591.4) and peaks at m/z 606.5 ($[M-H]^-$) arise either from UDP-GlcNAc (neutral mass 607.4) or from the hydrated (diol) form of UDP-2-acetamido-2,6-dideoxy-D-xylo-4-hexulose." In addition, we included the respective structures in the figure to make results clear.

F2: A true negative control would show the lack of radioactive species in the lane from no enzyme on the same phosphor screen. Showing the iodine stain is however a nice lane to show the Rf of C55P. This reviewer would be very curious to see where UDP-D-[14C]FucNAc runs in this system. What are the additional signals seen in the CapL lane?

We provide a new Figure 2, including the requested control on the same phosphor screen (lane 2; heat inactivated CapM) lacking additional signals and with reduced background. As expected there is no radioactive species migrating on TLC. Since the extracts are washed with water, residual unconverted UDP-D-[14C]FucNAc is fully eliminated from the BuOH extract and thus not visible on the TLC.

Usually, and if the BuOH extract were not washed, radiolabeled UDP-activated sugars migrate closely to the application spot. Because of the migration behavior we can exclude that signals seen in the (old) CapL lane correspond to UDP-activated sugars.

8. F2B: Results will be strengthened by high-res MS that has a mass accuracy within 0.5 ppm of the calculated mass. In addition MS/MS will help add evidence to the product/structure being

proposed. The structure of the polyisoprenoid tail in this structure is drawn incorrectly. Undecaprenyl has 7Z (Cis)-isoprene units and 3E (trans)-Isoprenene units with a terminal unit (=11). The structure drawn here has the wrong configuration and contains an extra CH₃ appendage in the Z-units.

We provide novel MS data (F2B) with improved mass accuracy of 2 ppm or lower. In addition, we now provide the mass of lipid II_{cap} adding additional support to the identity CP lipid precursors. The undecaprenyl structure has been corrected.

9. P8L14-15: There is no F.2C in the main text.

Fixed. Paragraph has been rewritten.

10. F4: The reviewer suggests that Cap5A be labeled as CapA1 as this is what the protein is referred to as in text. This would improve the readability of the manuscript.

We agree and have changed Cap5A to CapA1.

12. CapA1 and CapB2 are paralogs of each other. How similar in amino acid identity are the extended loops? A discussion should be presented as to potential reasons for differences in signal sensing and activities of this region at the molecular level.

We have now included an alignment of CapA1 and CapA2 (new Suppl Fig 9E) highlighting the loop region (blue). However, despite considerable sequence similarity of the loops (identities: 67/126(53%); positives: 93/126(73%), without structural data for both proteins (at best in complex with their signal molecules), such statements would be rather speculative.

12. Fig1. Attachment of the CP polymer to PG is missing a phosphate molecule in the peptidoglycan meshwork.

We have corrected Fig. 1 accordingly.

Reviewer #2 (Remarks to the Author):

This is an extensive and comprehensive study on capsule polysaccharide (CP) biosynthesis in *S. aureus*. The authors do not only succeed in reconstituting the major steps of this complex pathway *in vitro*, they also describe how the activity of several of the enzymes is controlled by different protein tyrosine kinases. The manuscript sheds new light on CP synthesis in *Staphylococcus aureus* by providing in depth biochemical characterization of key factors of the molecular machinery. Furthermore it underlines the biochemical findings by using *in vivo* data derived from analysis of capsule mutants. Especially interesting is the new finding that CapA seems to carry out an additional control function via a phosphodiesterase activity.

This study is an important contribution to our understanding of capsule polysaccharide, not only for *S. aureus*, but many bacterial pathogens using related pathways. However, the following points could be improved in the revised manuscript:

We thank the reviewer for the positive feedback and the comments that aided to improve the revised version of the manuscript.

1.Introduction: It is debated in the scientific community whether CP is a major virulence factor in *S. aureus* since major highly virulent strains (CA strains), such as MW2 and USA300, are deficient in CP. This issue could be mentioned and discussed better.

As suggested, we have now addressed this issue, particularly for USA300 strains, in the Introduction (page 3; lines 15-20). USA400 isolates (typified by strain MW2) are no longer prevalent in the United States, since they have been replaced by USA300 isolates. Moreover, not all USA400 strains are CP8 deficient (Montgomery CP et al, JID (2008) 198:561-570).

We have now included this aspect in the text (page 3, lines 12-20) and cite respective literature:

"In the case of Staphylococcus aureus, an important opportunistic pathogen [5], the expression of a polysaccharide capsule contributes substantially to the ability to cause invasive disease [6–8]. Serotype 5 and 8 capsular polysaccharide (CP5 and CP8) types are dominant among clinical isolates [7]. S. aureus USA300, which is prevalent in the United States, lacks a capsule due to several conserved mutations within the cap5 locus [9]. However, the majority of USA300-associated infection involved superficial wounds or abscesses [10], and USA300 isolates are not common outside of North America [11]. Among predominant methicillin-resistant S. aureus clones worldwide are the CP8+ lineages ST1, ST30, ST59, ST80, and ST239 and the CP5+ lineages ST5 and ST22."

2.CapC (CapC1 and CapC2) is believed to be a phosphatase, modulating the activity of the transmembrane adaptor CapA and BY-Kinase capB. Please explain why this protein was not included in the experiments and discuss its putative role.

We have now included CapC1 and CapC2 and provide biochemical evidence that both have phosphatase activity. In addition, we show that CapB kinase as well as target proteins (e.g. CapE) are dephosphorylated by CapC phosphatases (Suppl Fig 5D).

Page 7, line 12: The mass in Supplementary Figure 2 is indicated as 590.4. However in the text it is discussed as 591.4. Which is the correct mass? (UDP-FucNAc should have 591.08 according to this web tool (<http://www.sisweb.com/referenc/tools/exactmass.htm>))

We apologize for being imprecise. This point has also been raised by reviewer 1 (comment 6). We have clarified this in the text (page 7, lines 13-15): "...a sugar nucleotide species having a molecular mass of m/z 590.4 for the negatively charged molecule, consistent with the formation of UDP-D-FucNAc (Supplementary Figure 2)." and in the figure legend.

Line 16-18: Please explain why the negative control was not performed with boiled CapM or CapL?

We have now included a control with heat-inactivated CapM in the new Figure 2, to exemplarily demonstrate that the (radiolabeled) product bands do not derive from the protein purification/preparation.

Line 25-28: Why is the mass for the enzymatic product of CapM provided but not the mass of the following reaction by CapL?

We now provide the mass for the CapL reaction product lipid II_{cap}. We further show that lipid II_{cap} is further converted to lipid III_{cap} by CapI. However, despite extensive effort we were unable to obtain MS data for lipid III_{cap} due the limited availability of the product (requires at least 10 enzymatic steps) and even more importantly the fact that the products ionize poorly and do not fly in the MS. (see also answer to general comment of reviewer 1).

Page 8, line 24- Page 9, line 1: The authors should mention that tunicamycin inhibits CP synthesis *in vivo* as well.

Tunicamycin blocks peptidoglycan biosynthesis, i.e. MraY in lipid I biosynthesis at much lower concentrations, and thus CP synthesis inhibition *in vivo* does appear less relevant, since CP is attached to peptidoglycan. We have included IC50 values and compare those of CapM to polytopic GTs, such as MraY or TarO (page 9, lines 13-16) to make this clearer.

Page 9, line 22 – Page 10, line 4: which CapA and B homologue is "CapAB"? From the figure legend it can be concluded that CapA1B1-fusion was used. This should be clarified in the text.

CapA1B1 has been used. This has been clarified in the text, figures and throughout the manuscript.

Line 22-25: It is not clear from the text, why the authors tested only proteins CapD CapN CaPM and CapE for phosphorylation by CapAB. CapF and CapG are discussed to be no target of CapAB-phosphorylation but this does not seem to be shown. Please clarify.

We now show that CapF, CapG and CapL are not targets of CapA1B1 (new Suppl Fig 5c).

Page 10, line 21-25: The authors describe the structural features of CapE in detail - this could be made more clear by including it in Figure 3.

The CapE structure is published, and this is cited; the color code used in Figure 3F matches those used in Miyafusa et al.2013. Relevant features, such as the active site and the MK motif as well as the "mobile latch" are highlighted in the Figure and explained in the figure legend and main text to make this clearer.

Page 11, line 12- Page 12, line 4: The text Refers to the enzyme as CapA1- however in Figure 4 A+B it is indicated as Cap5A. Please clarify.

We have changed Cap5A (gene annotation) to CapA1 throughout the text and in the figures and legends.

Page 12, line 9-10: Complementation of mutant CapA1 and CapB1 could be included in Figure 5C.

Figures have been rearranged and updated. We show complementation of the *capA1* deficient *S. aureus* MW2 (Fig 4D) and the *capB1* knock-out in *S. aureus* Newman (new Suppl Fig 4C). In addition, the Δ pknB mutant was complemented *in trans* (new Fig 6E).

Page 14, line 1-2: It is not clear from the figure (Fig 6 A) why PBP2 is catalytically active.

This has been clarified in the text and figure legend, e.g. Page 16, line 5-7: The resulting polymerized PBP reaction product is not extracted from the reaction mixture, since it lacks the bactoprenol tail, thus lipid bands vanish from the TLC. We further included a moenomycin control, to show that the conversion of residual lipid II and the Lipid II_{PG}-CP product by PBP2 is inhibited (bands are visible again) (now Fig 5B).

Importantly, we now further show that PBP2 is able to convert the ultimate peptidoglycan precursor lipid II to which the CP trisaccharide has been attached (Fig 5D).

Page 15, line 5-7, Figure 7C: The complemented PknB mutant could be included
The Δ pknB mutant was complemented *in trans* (new Fig 6E).

Discussion: The first paragraphs repeat data and conclusions from the results section and could be considerably shortened.

The text has been optimized accordingly, limiting repetition of results in the discussion (and vice versa).

Page 17, line 20-24: How can an enhancing effect of CapA1 on LcpC be explained on the molecular level?

Without structural data for both (membrane) proteins (in complex) this is difficult to predict. We propose that CapA1 and LcpC interact by forming a stable or transient complex, thereby modulating LcpC mediated attachment of CP to the cell wall precursor lipid II. We rephrased the sentence accordingly, to make this clearer (page 19, line 2-6). A similar situation has been proposed for the analogous CpsC and CpsA proteins in *S. agalactiae*, which is discussed and cited.

“Importantly, CP lipid precursor cleavage and transfer of the phospho-sugar moiety were found to be enhanced in the presence of CapA1, suggesting that the transmembrane activator cooperates with LcpC by forming an interaction complex, thereby modulating the attachment of CP to the cell wall precursor. Of note, Toniolo et al. (2015) suggested an equivalent role for the CapA1 homolog CpsC in CP biosynthesis of Streptococcus agalactiae.”

Page 18, Line 6: It is not clear from the data why PBP2 is active

See comment above.

Figure 1B: What does the small arrow or double arrow in the cartoon enzymes mean? Please describe in the Figure legend.

Small arrows in the cartoon enzymes indicate synthesis direction. The double arrow in CapM indicates reversibility of the reaction, as described for numerous GTs. This is now explained in the Fig legend.

Figure 3B: Why are there two bands for CapM and CapE (Protein degradation products? Issue is not addressed.)

We have optimized the Figure labeling, to make clear that the upper band corresponds to autophosphorylated CapA1B1_{fus}. The lower bands correspond to the trans-phosphorylated CapE and CapM, respectively.

Figure 4 A and B: Cap5A should be called CapA1 as in the corresponding text.
Fixed in the entire manuscript.

Figure 6: What do we see for PBP2?

See comment above.

Reviewer #3 (Remarks to the Author):

Rausch and colleagues use a biochemical approach to study capsular biosynthesis of *Staphylococcus aureus*. Using purified Cap proteins, sugar nucleotide precursors and undecaprenyl-phosphate, the authors demonstrate CapM catalyzes the synthesis of lipid I/cap (C55-PP-FucNAc), that CapAB phosphorylate CapM and CapE to control capsular synthesis, that CapA exerts phosphodiesterase activity on lipid I/cap, that LcpC links lipid I/cap to lipid II/peptidoglycan and that PBP2 polymerizes not only lipid II/peptidoglycan but also lipid II/peptidoglycan/FucNAc, and that the threonine/serine kinase PknB acts on CapM to impact capsular polysaccharide synthesis.

General comments

The paper is well written, interesting to a large audience, and provides new biochemical insights on

capsular polysaccharide synthesis. This said, the paper seems also odd in that it describes the CapM initiating reaction which leads to lipid I/cap but not the remainder of the biosynthetic pathway.

We have now reconstituted the entire biosynthetic pathway, resulting in the synthesis of the ultimate full-length CP lipid intermediate lipid III_{cap}.

To the best of our knowledge there is no other paper published that describes the reconstitution of an entire cell wall biosynthetic pathway using full-length purified membrane proteins and substrates.

To this, the authors add the study of CapAB-mediated regulation of CP synthesis, which involves CapM and CapE. The studies on LcpC are even further removed, especially as LcpC is thought to use lipid III/cap as a substrate, not lipid I/cap. With this in mind, the question remains whether LcpC actually catalyzes the addition of lipid III/cap to lipid II or to peptidoglycan. I am not convinced that the observed LcpC synthesis of lipid I/cap with lipid II/PG is meaningful.

We have now included additional data, showing that LcpC uses lipid III_{cap} and efficiently attaches the trisaccharide to the acceptor substrate lipid II_{PG} (Fig 5D). In addition, and with regard to the membrane localization of both, substrates and enzymes, and the interdependency and intimate connection of the enzymatic machineries, lipid II_{PG} is the most plausible acceptor. Furthermore, we show that PBP2 is able to polymerize the LcpC product Lipid II_{PG}-CP and that this reaction is inhibited by moenomycin.

We also refer to the recently published study of Schaefer et al (Nature Chem Biol, 2018) on LcpA and B (Page 19, line 11-16), stating that “nascent PG” is the acceptor of WTA. However, it has to be taken into account that this study used soluble, shortened precursor mimics (to allow for MS analysis), proteins lacking the TM domains and an acceptor substrate (“nascent PG”) that has so far not been shown to even exist. This may explain why hydrolase activity is not observed for these WTA substrates.

Specific comments

1. The Abstract does not capture the experimental work described in this paper.

We have modified the abstract and believe that it now captures all relevant information.

2. What is the source of C55-P and what is its purity?

C55-P is commercially available (purity > 95%). This information is now included in the material and method section (page 29, line 16).

3. The Discussion is excessively long, boring and at times redundant with the Results section.

The text has been optimized, limiting repetition of results.

4. If the authors want to keep the LcpC data in the paper, I would advise that the shortcomings of the experimental design are explained in detail.

We suppose that the reviewer refers to the use of lipid I_{cap} as the substrate for LcpC. We have addressed this issue and now include data on the attachment of lipid II_{cap} and lipid III_{cap} (see general comment). Furthermore, we show that PBP2 is able to polymerize the LcpC synthesis product and that this reaction is inhibited by moenomycin.

5. The product of the LcpC catalyzed reaction, C55-PP-PG-FucNAc was not characterized by mass spectrometry. How can the authors be sure the TLC signal represents this product?

As outlined above we now include attachment of all three capsular intermediates to lipid II_{PG}. Consistently, the respective reaction products are characterized by an altered migration behavior on TLC, resulting from the attachment of either one, two or three sugar residues. Of note, the acceptor substrate lipid II_{PG} is radiolabeled and so are the products.

Mass spectrometry proved impossible for the Lcp products, due the long bactoprenyl lipid tail. We already observed that much higher amounts of Lipid II_{cap} were required for MS analysis compared to lipid I_{cap}, suggesting that an increase in sugar residues attached to C55-P impacts on ionization. This was also observed for the LcpC reaction product (see also general comment to comment of reviewer 1). Corroborating, Schaefer et al used shortened (C20), soluble WTA precursors to study attachment of WTA by LcpA and LcpB, which enabled MS analysis (see general comment).

Reviewer #4 (Remarks to the Author):

In this manuscript, authors have performed excellent biochemistry to reconstitute capsule biosynthesis *in vitro*. In the process they established biochemical assay for CapM enzymes, paving the way for investigating the regulation of its activity. To investigate the regulation of CP synthesis, authors focused on bacterial tyrosine kinases and the eukaryotic serine/threonine kinase PknB. Authors show that CapB1 is a functional kinase and identify CapM and CapE to be direct targets of CapAB chimera. Further work elucidated CapA1 to be a dual-functional protein with phosphodiesterase and kinase activator activity. Interestingly, despite functional redundancy *in vitro*, CapA2 and CapB2 do not play any role in modulating CP production, indicating specific CP independent function for CapB2 kinase *in vivo*. Overall I believe the manuscript has some very interesting and novel data. *In vitro* experiments are very well performed. I find the regulation mediated through phosphorylation on Tyr residues and Thr residues quite intriguing. I also find the cross-phosphorylation of BY kinase by ESTK quite interesting. However, these conclusion are solely based on *in vitro* experiments. I believe that *in vivo* evidence is critical for demonstrating phosphorylation-mediated regulation and the role of cross-phosphorylation in modulating Capsule assembly. My specific comments are appended below.

We appreciate the reviewer's comment. Thank you.

We disagree however on the statement that our conclusions are solely based on *in vitro* experiments. We provide solid *in vivo* data that in a CapA1 mutant (but not a CapA2 mutant) CP formation is strongly reduced. In addition, we show that a PknB deletion mutant CP production is strongly elevated. We now show that these phenotypes are complemented by expressing *capA1* and *pknB* *in trans*. In combination with our *in vitro* data the conclusions are fully supported, since we observed that PknB mediated phosphorylation of CapM or CapB1 decreases the respective enzyme activities and that CapAB1 mediated phosphorylation enhances enzymatic activity of key biosynthetic enzymes. In the case of CapB1 we clearly show that this effect results from PknB cross-phosphorylation and this effect is reflected by the *in vivo* studies and now further supported by respective phosphomimetic and phosphoablative mutants.

Besides these *in vivo* data, we particularly focused on the isolated *in vitro* systems for various reasons. Importantly, as this study shows the different cell wall biosynthetic pathways are tightly interlinked and this is underlying a likewise high degree of regulation and cross-regulation (that is by far not completely understood). Furthermore, the investigated regulatory systems (i.e. CapAB and PknB) are shown to antagonistically affect each other. Moreover, additional modulation comes from the transcriptional level and the interference with various other regulatory systems in the cell. By manipulating the cell by knock out or overexpression of proteins *in trans*, compensatory side effects cannot be excluded. Importantly, this is most relevant if we do not even understand the functions of these proteins on the molecular level, which is the main focus of this paper. As shown

here CapA1 has such crucial function that goes beyond what was previously expected. And this is also true for some of the other CP biosynthesis proteins, which may have for example scaffolding activities.

Detailed comments below.

Major Comments:

It is not clear to the reviewer as to how authors have arrived at Tyr157 as the primary phosphorylation site. There is no mass spec data presented either in the main text or in the supplementary to support this conclusion. This data requires support from mass spec data and *in vitro* kinase assays with the wild type and mutant proteins.

An *in silico* phosphosite prediction using NetPhos Server was initially used, which is common practice. This software identified only 2 Tyr residues (Tyr75, Tyr157) as potential phosphorylation sites on CapM. This information has been added to the text (Page 10, line 24-25). Only one of these residues (Tyr157) is highly conserved in homologous glycosyltransferases. Furthermore, *in vitro* kinase assays using mutant CapM proteins (Tyr75Phe or Tyr157Phe) unequivocally identified Tyr157 as the relevant phosphosite. These data are now further supported by a phosphomimetic CapM variant (Tyr157Glu), determination of kinetic parameters as well as structural modelling of CapM.

Based on the *in vitro* experiments authors suggest that Tyr157 on CapM and four tyrosine residues on CapE are target phosphorylation sites. Manuscript lacks *in vivo* evidence for phosphorylation of substrates (CapM by PknB and CapB; CapE by CapB and CapAB by PknB). It is necessary to show that CapM, CapE and CapB1 (on Thr residues) are indeed phosphorylated *in vivo*. It is also important to identify the *in vivo* phosphorylation sites with the help of mass spec and determine the stoichiometry of phosphorylation at different locations (different Thr and Tyr residues).

As described for CapM, phosphorylation of target proteins by either PknB or CapAB1 has been extensively characterized using diverse mutant proteins.

In vivo evidence (providing at least additional support) as requested, requires extremely extensive phosphoproteome analysis of diverse mutant strains and combinations at different growth phases, which is a tremendous effort, particularly in *S. aureus*. In our opinion this goes far beyond the scope of a single paper (or even more publications). See also general comment.

It is not clear to the reviewer how authors have performed the experiment presented in Fig 7E. I did not see any experiment showing phosphorylation of CapAB by PknB, nor did I see any description of the experiment either in methods or in the legend. How did they differentiate between PknB mediated phosphorylation of CapAB and autophosphorylation of CapAB? Why the experiment was not performed with phospho-ablative and phospho-mimetic mutants to eliminate the ambiguity?

We show PknB-mediated phosphorylation of CapB1, supported by mass spec (Suppl Fig. 12), since one cannot differentiate Tyr autophosphorylation and Thr transphosphorylation by PknB of the CapAB fusion protein. PknB mediated phosphorylation interferes with CapB1 autophosphorylation, resulting in reduced phosphorylation of CapAB1 compared to the control (no PknB), although the inhibitory effect should be more pronounced (sum of Tyr and Thr phosphorylation vs. Tyr autophosphorylation). To evaluate the impact of PknB-mediated phosphorylation on CapAB autophosphorylation, phosphomimetic (T8E) and -ablative (T8A) proteins are now included (Supp. Fig 12), as recommended, to make this even more clear.

Generation of deltaCapM mutant and complementation experiments with the wild type and mutants and their relative impact on CP production would be necessary to investigate regulatory aspects. Similarly complementation of deltaCapB1 mutant with the wild type CapB1 and PknB target site mutants would be necessary to show a role for cross-phosphorylation in CP production.

See answers to general comment and comment 2. As outlined above complementation of deltaCapB1 is now included, restoring CP production in vivo.

Minor Comments:

There is certain amount of redundancy between results and discussion part. Specific example would be Page 10 (lines 19-26) and page 11 (lines 1-2). Almost the same contents are rewritten in the Page 16 (lines 7 to 17). Authors need to rework such parts to make the manuscript more concise.

We have optimized these parts accordingly.

Loading controls are required for both Figure 3A and 3B. Autophosphorylation and trans-phosphorylation of CapM and CapE bands should be indicated for clarity. In the text authors state that CapD, CapN, CapG and CapF are not phosphorylated. However in the figure authors have neither performed the reaction with CapG nor CapF.

Figure 3 has been optimized and autophosphorylation and trans-phosphorylation is now indicated. Loading controls are now provided in Suppl. Fig. 5 A and B. We further now show that CapF, CapG and CapL are no targets of CapAB (Suppl Fig 5C).

Figure 7D requires CapB1 and CapB2 alone controls.

Figure 7D (new Fig 6C) has been changed, now showing CapB1 only. The requested CapB1 alone control is already shown in Fig. 3A.

In Figure 4A, authors label CapA1 as Cap5A. Though elsewhere in the manuscript, authors do state that Cap5a and CapA1 are the same, for the ease of understanding labeling should be kept consistent.

Fixed throughout the text.

Sincerely,

Tanja Schneider

REVIEWERS' COMMENTS:

Reviewer #1 (Remarks to the Author):

The authors have responded admirably and effectively to all of the concerns raised in my previous assessment. I have no further comments for the authors.

Reviewer #2 (Remarks to the Author):

All my concerns were sufficiently addressed. I recommend publication.

Reviewer #4 (Remarks to the Author):

I have gone through the revised manuscript. I appreciate the efforts made by the authors to address the questions to the best of their ability and feasibility. Authors have addressed most of the questions by performing additional experiments and providing appropriate controls and explanations.

I would like authors to address the following concern:

Based on the prediction, authors pursued Y75 and Y157 as the possible target sites for CapA1B1 phosphorylation (Fig 3D). They show that CapM Y157F mutant fails to show increased activity (Sup Fig 6A). Thus they conclude that Y157 is the primary site for CapA1B1 and it regulates the activity of CapM. I would like authors to perform an in vitro kinase assays with CapM, CapM Y75F, CapM Y157F and CapM Y75+157F. If these are indeed the sites, CapA1B1 would no longer phosphorylate CapM Y75+157F mutant.

Response to REVIEWERS' COMMENTS:

Reviewer #1 (Remarks to the Author):

The authors have responded admirably and effectively to all of the concerns raised in my previous assessment. I have no further comments for the authors.

Reviewer #2 (Remarks to the Author):

All my concerns were sufficiently addressed. I recommend publication.

Reviewer #4 (Remarks to the Author):

I have gone through the revised manuscript. I appreciate the efforts made by the authors to address the questions to the best of their ability and feasibility. Authors have addressed most of the questions by performing additional experiments and providing appropriate controls and explanations.

I would like authors to address the following concern:

Based on the prediction, authors pursued Y75 and Y157 as the possible target sites for CapA1B1 phosphorylation (Fig 3D). They show that CapM Y157F mutant fails to show increased activity (Sup Fig 6A). Thus they conclude that Y157 is the primary site for CapA1B1 and it regulates the activity of CapM. I would like authors to perform an in vitro kinase assays with CapM, CapM Y75F, CapM Y157F and CapM Y75+157F. If these are indeed the sites, CapA1B1 would no longer phosphorylate CapM Y75+157F mutant.

We thank all reviewers for their time and input which helped to improve the manuscript.

To provide further evidence that Y157 is the primary phosphorylation site for CapA1B1 we have included an in vitro kinase assay (new Suppl Fig 6A) showing that in contrast to CapM_Y75F which is still phosphorylated by CapA1B1, a CapM_Y157F mutant showed no phosphorylation by CapA1B1.

We have updated the text accordingly (page 10, line 25): The highly conserved Tyr157 appears to be the primary phosphorylation site in CapM, since tyrosine phosphorylation and the concomitant stimulatory effect on the catalytic activity was completely abolished in the CapM_Y157F mutant (Supplementary Figure 6A, B).